# Higher stability of novel live-attenuated oral poliovirus type 2 (nOPV2) despite the emergence of a neurovirulent double recombinant strain in Uganda

The novel oral poliovirus vaccine type 2 (nOPV2) was developed to reduce the risk of circulating vaccine-derived poliovirus outbreaks by incorporating genetic modifications to enhance genetic stability and reduce reversion to virulence while retaining protection. Here we report the characterization of 231 nOPV2 isolates from Uganda during a 1-year period following nOPV2 use. Whole-genome sequencing revealed that most isolates retained nOPV2's genetic modifications, with limited mutations in the VP1 region indicating no relevant virus transmission. However, a double recombinant strain identified in a sewage sample lost all key nOPV2 modifications through recombination with enterovirus C strains upstream and downstream of the capsid coding region. This resulted in high neurovirulence comparable to that of wild-type 2 poliovirus. Despite this, the strain did not spread widely, probably due to high vaccination coverage. These findings underscore the enhanced genetic stability of nOPV2 and its reduced risk of reversion compared with Sabin monovalent OPV2 (mOPV2), while highlighting the importance of surveillance to detect rare recombination events. Continued use of nOPV2 and inactivated polio vaccine, combined with robust immunization and monitoring, remains essential for achieving and sustaining global polio eradication.

The live-attenuated oral poliovirus vaccine (OPV) has proven to be a very effective tool to protect against poliomyelitis and interrupt poliovirus (PV) transmission, being the primary strategy driving the Global Polio Eradication Initiative (GPEI) for more than 60 years. This has helped to reduce wild PV transmission to the brink of extinction, with wild PV types 2 and 3 being certified eradicated and wild PV type 1 cornered in the remaining two endemic countries Afghanistan and Pakistan[1,2]. However, the rare ability of OPV strains to regain virulence and establish circulating vaccine-derived poliovirus (cVDPV) outbreaks in areas with persistently low immunity poses a substantial risk to eradication efforts. This risk is particularly pronounced for PV type 2 (PV2), as wild PV type 2 was declared eradicated in 2015, and immunity gaps following

the global cessation of OPV type 2 use from routine immunization have allowed cVDPV2 outbreaks to emerge[3,4].

To address these challenges, a modified version of the vaccine, designated novel OPV type 2 vaccine strain (nOPV2) was developed that is genetically more stable and less likely to regain virulence than the original Sabin OPV2 strain[5]. This was achieved by introducing genetic modifications that include stabilizing the 5′ non-translated region (5′NTR) to preserve attenuation determinants, relocating the *cis*-acting replication element (CRE) from the 2C coding region to the 5′NTR (CRE5) to reduce recombination risk, and introducing mutations in the 3D polymerase (Rec1 and HiFi) to further reduce recombination frequency and improve replication fidelity. In addition, nOPV2 contains three unique silent

e-mail: Javier.Martin@mhra.gov.uk

mutations in the VP4-VP2 capsid coding region incorporated during the nOPV2 cloning process. nOPV2 demonstrated robust immunogenicity and safety profiles in clinical trials conducted between 2017–2019, effectively inducing strong immune responses while minimizing the risks associated with viral reversion[6–8]. For this reason, the World Health Organization (WHO) in 2020, issued an Emergency Use Listing (EUL) recommendation for the use of nOPV2 in response to cVDPV2 outbreaks[9,10].

A key requirement of the EUL framework was the molecular characterization of nOPV2 viruses isolated from patients diagnosed with acute flaccid paralysis (AFP) and from environmental surveillance (ES) to monitor genetic stability and to detect mutations potentially impacting neurovirulence and/or transmissibility. Data generated from whole-genome sequencing of nOPV2 virus isolates from AFP patients and ES were reviewed by the genetic characterization subgroup of the nOPV2 working group within the GPEI[11]. During the initial use phase (March–October 2021), ~111 million nOPV2 doses were administered in seven countries, and whole-genome sequencing of 251 isolates confirmed the genetic stability of nOPV2 compared with the original Sabin monovalent OPV2 (mOPV2) based on the preservation of the main attenuation site domain V in the 5′NTR in all nOPV2 isolates[10,12]. This informed the subsequent Strategic Advisory Group of Experts on Immunization (SAGE) and WHO decision to grant approval for wider use of nOPV2 under the EUL framework[9].

In this context, Uganda detected two cVDPV2 isolates from ES samples collected from the Lubigi sewage treatment plant, located in the capital city of Uganda, Kampala, on 1 June and 2 November 2021[13]. These isolates were genetically linked to cVDPV2 isolates previously identified in Sudan in 2020, belonging to the CHA-NDJ-1 emergence first detected in the Republic of Chad in 2019. Following the detection of these cVDPV2 isolates, the Ministry of Health declared a public health emergency of international concern on 12 August 2021. A risk assessment concluded that due to the high virologic risk, gaps identified in AFP surveillance, low coverage of inactivated polio vaccine for the 2 years following its introduction, high population density in the Kampala metropolitan area, and high risk of importation of cVDPV2 from neighbouring countries, Uganda should be classified as high-risk for polio re-emergence. In response, two nationwide nOPV2 campaigns were conducted in Uganda between 14–21 January and from 4 November 2022 until 28 February 2023[14], immunizing ~9.8 million and 11 million children, respectively. The second round was conducted in 3 phases: phase 1 from 4–28 November 2022; phase 2 for four of the five Ebola-affected districts (Kassanda, Mukono, Mubende and Wakiso) from 20–27 January; and phase 3 for Kampala district from 20–28 February 2023. Despite challenges with the Ebola outbreak, ~1.6 million children were immunized in the five high-risk districts.

This study reports the genetic characterization of 231 nOPV2 isolates from Uganda following the vaccination campaigns, 147 from sewage samples and 84 from stools of AFP cases and their contacts. The primary aim was to further evaluate the genetic stability of nOPV2 and identify genomic changes potentially influencing genetic stability and other viral properties. A double recombinant nOPV2 strain, which had replaced all nOPV2-specific sequences upstream and downstream of the capsid coding region with those from enterovirus (EV)-C strains resulting in the loss of all major nOPV2 genetically engineered modifications, was identified, raising concerns about its potential to regain virulence.

The findings are discussed in the context of further evaluating nOPV2 as an improved vaccine for the GPEI to interrupt cVDPV2 circulation while minimizing the risk of generating new cVDPV2 outbreaks, compared with Sabin mOPV2, which has been used for over six decades.

## Results

### Isolation of PV2
Between January 2022 and March 2023, a total of 2,773 stool and 205 ES specimens were processed in the WHO-accredited National Polio

**Table 1 | Classification of nOPV2 isolates from Uganda based on genome composition**

| Category | Total | AFP | ES | Type of isolate |
|---|---|---|---|---|
| 1 | 0 | 0 | 0 | VDPV2-n; double recombinant |
| 2 | 3 | 0 | 3 | nOPV2-like (nOPV2-L); double recombinant |
| 3 | 0 | 0 | 0 | VDPV2-n; single recombinant upstream KO CRE |
| 4 | 0 | 0 | 0 | nOPV2-L; single recombinant upstream KO CRE |
| 5 | 0 | 0 | 0 | VDPV2-n; single recombinant downstream KO CRE |
| 6 | 6 | 1 | 5 | nOPV2-L; single recombinant downstream KO CRE |
| 7 | 0 | 0 | 0 | VDPV2; non-recombinant |
| 8 | 217 | 81 | 136 | nOPV2-L; non-recombinant |
| 9 | 5 | 2 | 3 | nOPV2-L; no mutations |
| **Total** | **231** | **84** | **147** | |

Laboratory in Uganda. Of these, 100 (48.8%) sewage samples and 123 (4.4%) stool samples yielded PV isolates in cell culture. A total of 147 PV isolates from ES samples and 84 from stool samples were identified as PV2 by intratypic differentiation (ITD). All PV2 isolates (except 3 isolates from one sewage sample collected on 15 February 2022 from Kisenyi sewage ponds in Kabarole district) were confirmed as nOPV2+ in the ITD assay, indicating preservation of the nOPV2-relocated CRE5 domain in the 5′NTR as primers for this ITD reaction targets this region. The time interval between nOPV2 vaccination campaigns and the collection of PV2-positive samples ranged from 2 to 46 days for AFP samples (mean = 16.6, median = 13) and from 5 to 109 days for ES samples (mean = 30.7, median = 32).

### Whole-genome sequence analysis of PV2 isolates and nOPV2 category classification
Whole-genome sequences of the 231 PV2 isolates were determined using Illumina deep sequencing of overlapping tiled PCR products. Sequences were aligned and compared with the nOPV2 vaccine strain (GenBank ID: MZ245455) to confirm nOPV2 origin and identify mutations or recombination events. All isolates were confirmed to be of nOPV2 origin, retaining some or all the nOPV2-specific genetic modifications.

Isolates were classified into nOPV2 categories 1–9 on the basis of mutation and recombination profiles[12]. This ranking matches with the risk of emergence of pathogenic strains from category 9 (no mutation) to category 1 (double recombinant VDPV2 strains). As shown in Table 1, 5 isolates (2.2%) showed no changes from nOPV2 (category 9), while the majority (n = 217; 93.9%) were classified as category 8, exhibiting no reversion or recombination and limited VP1 substitutions (0–5). Six isolates (2.6%) were classified as category 6, showing limited VP1 substitutions (0–5) but containing single recombination events with sequences derived from nOPV2 and Sabin 1 (n = 2) or unidentified species C EVs (n = 4). These recombination events preserved the nOPV2 KO CRE motif in the 2C protein coding region but lost one or both nOPV2 HiFi and Rec1 3D polymerase mutations (Extended Data Fig. 1). Finally, the three isolates from the Kisenyi Sewage Ponds site, which showed a PV2+/nOPV2− ITD result, exhibited a double recombinant structure, replacing all nOPV2 genetic modifications upstream and downstream of the capsid coding region with sequences from unidentified species C EVs (Fig. 1). The replacement of the nOPV2 5′NTR explains the nOPV2− ITD result. The nOPV2 origin of these three isolates was confirmed by the presence of the three unique silent mutations in the capsid-coding region (C814A, C817U and A1375U) that were incorporated during the

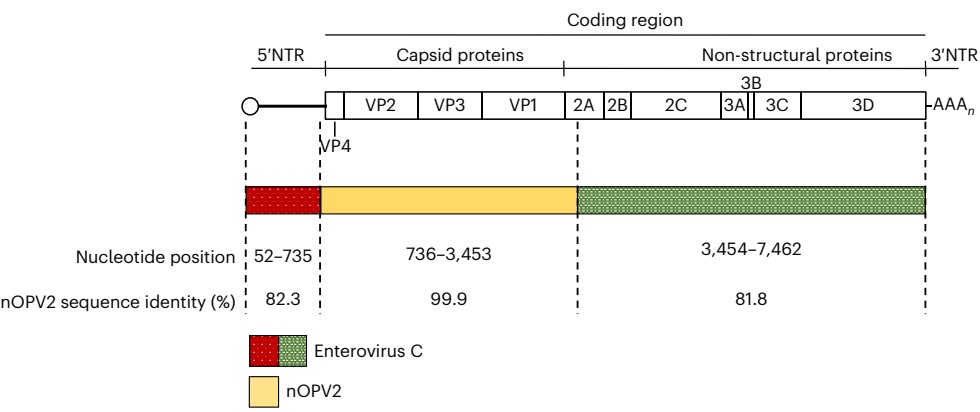

**Fig. 1 | Genomic structure of double recombinant nOPV2 strain from Kabarole (Uganda).** Schematic representation of double recombinant nOPV2 strain from Kabarole (Uganda) with sequences derived from the nOPV2 vaccine strain and unidentified enterovirus C strains.

nOPV2 cloning process. These viruses were classified as category 2 as they show a double recombinant structure and <5 VP1 nt changes (*n* = 1–2) from the nOPV2 vaccine strain.

No VDPV2 strains associated with nOPV2 use (VDPV2-n) were detected in Uganda, as the maximum number of VP1 nucleotide changes from the nOPV2 vaccine strain was 5, below the threshold that separates vaccine-like strains and VDPV2s (6 nt changes). Importantly, no cVDPV2 isolates linked to the viruses found in Lubigi ES samples from July and November 2021 have been detected in any stool or ES samples from Uganda ever since.

### Identification of mutations in nOPV2 isolates

The number of mutations identified in nOPV2 isolates as a function of time after the corresponding supplementary immunization activity (SIA) campaigns is represented in Fig. 2. A simple linear regression analysis was used to estimate the rate of sequence evolution. On the basis of this analysis, a sequence evolution rate was estimated, giving very similar values of 1.36% and 1.30% mutations per site per year using data from isolates from AFP and ES samples, respectively (Fig. 2), as expected for poliovirus evolution.

Notably, the primary attenuation site (stabilized domain V in the 5′NTR) remained unchanged in all isolates except the three double recombinants, which had replaced the entire 5′NTR. Two isolates showed mutations in this region: one with C547U, weakening a stem region and potentially increasing attenuation, and another with A566G in a loop region, with no expected effect on neurovirulence.

The frequencies and possible effect of key mutations observed in nOPV2 isolates are shown in Fig. 3 and Table 2, respectively. The most frequent mutations were noted at nucleotide positions that have been shown or inferred to individually decrease attenuation slightly, located at CRE5, domain II and domain IV in the 5′NTR and VP1 amino acids 143 and 171. We found 36 different sequence combinations at these positions, including 24 isolates with no mutations and 207 isolates containing 1–5 mutations per isolate (Extended Data Table 2).

### Genetic characterization of the double recombinant nOPV2 strain from Kabarole

The three double recombinant isolates from Kabarole exhibited nearly identical whole-genome sequences, with sequence identity to nOPV2 varying across the genome: 82.3% in the 5′NTR, 99.9% in the capsid coding region and 81.8% in the non-structural coding region (Fig. 1), demonstrating a double recombinant structure. In addition, all three isolates showed reversion at secondary attenuation site VP1–143. Phylogenetic analysis revealed no close genetic links of this strain to any other enterovirus strain in Genbank in the 5′NTR and the non-structural coding region.

In addition to Flinders Technology Associates (FTA) cards, all original sewage concentrates and recovered virus isolates from the Kisenyi ES site were sent to the Medicines and Healthcare products Regulatory Agency (MHRA, UK) for further analysis. Whole-genome sequence analysis of infectious poliovirus isolates recovered from RNA from FTA cards by transfection and from sewage concentrates by cell culture infection corroborated the initial observations. Two additional double recombinant isolates, identical or nearly identical to the previous three, were identified in the ES sample collected on 15 February 2022 from the Kisenyi site. In contrast, all other sequenced nOPV2 isolates from the same site, collected before and after 15 February 2022, exhibited a non-recombinant genomic structure and were genetically distinct from the double recombinant strain.

### Neurovirulence assessment of the double recombinant nOPV2 strain from Kabarole in transgenic mice

The neurovirulence of the double recombinant strain was assessed using transgenic mice expressing the human poliovirus receptor. As shown in Table 3, the dose required to cause paralysis in 50% of transgenic animals (50% paralytic dose, $PD_{50}$) for the Uganda strain was 1.1 (95% confidence interval (CI) 0.6–1.6) $\log_{10}(CCID_{50}$, the 50% cell culture infectious dose), substantially lower than that of nOPV2 (>8.4) and the original Sabin 2 vaccine (5.8; 95% CI 5.1–6.4) and closer to the wild PV type 2 strain MEF-1 (0.7; 95% CI 0.2–1.2) and mOPV2 with the revertant 481G mutation (1.8; 95% CI 1.4–2.2), indicating a neurovirulent phenotype as expected from the observed revertant genotype (Table 3).

### Detection of poliovirus transmission following SIA campaigns

No evidence of prolonged transmission of any nOPV2 strain was observed, including the double recombinant virus from Kisenyi, as VP1 nucleotide changes in nOPV2 isolates ranged from 0 to 5, below the threshold for VDPV2 classification. Of the 84 children sampled, only 7 were confirmed to have received nOPV2. Given the dynamics of household and community transmission, some nOPV2 isolates detected in stool were probably attributable to secondary spread. Importantly, sequence analysis showed no evidence of sustained transmission, and the maximum observed interval between vaccination and detection of a positive stool sample was 46 days, supporting the conclusion that prolonged shedding did not occur. However, two possible short-term transmission events were identified by sequential viruses from the same ES site showing close genetic links (Extended Data Fig. 2). The first cluster involved isolates from one site collected over 2 weeks; they shared 9 mutations from nOPV2, two in the VP1. The second cluster included isolates from another site collected over 46 days; these viruses shared 7 mutations from the vaccine strain, two in the VP1 region. None of the clusters involved recombination. In any case, these events represent

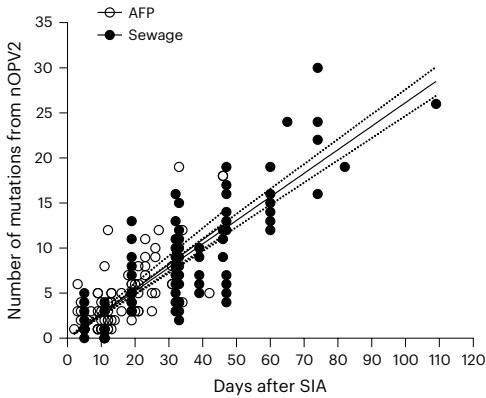

**Fig. 2 | Number of mutations from nOPV2 vaccine strain found in nOPV2 isolates.** The nOPV2 vaccine strain (GenBank ID MZ245455) was found in nOPV2 isolates from AFP (white circles) and ES samples (black circles). Data were adjusted to a linear function for the accumulation of nucleotide substitutions ($Y = 0.2733 \times X$, $Y = 0.2624 \times X$). Solid line shows best-fit simple linear regression of number of mutations versus days after SIA; dotted lines indicate the 95% confidence interval of the regression. Source data are provided.

short periods of time and can also be explained by shedding by the same individual/household.

### Prevalence of non-polio enterovirus serotypes in wastewater samples from Kabarole

A retrospective analysis of wastewater samples from the Kisenyi site (January–April 2022) identified 64 unique EV strains from 33 serotypes across species A ($n = 7$), B ($n = 16$) and C ($n = 10$). Species C EVs were the most prevalent, with multiple strains from serotypes known to recombine with PV, including CV-A11 ($n = 2$), CV-A13 ($n = 9$), CV-A17 ($n = 3$), CV-A20 ($n = 5$) and CV-A24 ($n = 2$) representing 27.27% of the total sequence reads (Extended Data Figs. 3 and 4). PV2 was detected in ES samples from 18 January, 1 February and 15 February 2022, consistent with routine virus isolation results.

### Discussion

Uganda implemented nOPV2-based targeted immunization campaigns under the WHO EUL framework to interrupt cVDPV2 transmission after the detection of two genetically linked cVDPV2 isolates from an ES site in Lubigi in July and November 2021[14]. As of 30 January 2025, no additional cVDPV2 isolates related to this emergence have been identified in Uganda, suggesting nOPV2 was effective in interrupting transmission.

The whole-genome sequences of 231 nOPV2 isolates were analysed over a 1-year period of enhanced surveillance following vaccination. Remarkably, as observed in nOPV2 isolates from the nOPV2 initial use phase (March–October 2021[12]), no mutations affecting RNA secondary structure base-pairing were detected in the nOPV2 primary attenuation site domain V in the 5′NTR in any of the isolates, unlike mOPV2, which typically reverts at this site in nearly all vaccinees within days[7,8].

Nevertheless, we identified molecular pathways by which nOPV2 can lose attenuation properties, including the accumulation of single mutations that alone reduce attenuation slightly and/or recombination with co-circulating enterovirus C strains. Mutations strengthening base pairing in RNA secondary structure domains in the 5′NTR, which are known or predicted to slightly reduce attenuation when present individually, together with capsid mutations affecting antigenicity and attenuation[8,12], were observed. However, based on the analysis of similar viruses shed by vaccinees in clinical trials[7,8] and laboratory studies of molecular clones containing increasing numbers of the observed mutations (A.M. et al., manuscript in preparation), few, if any, of these mutation combinations identified in nOPV2 isolates would restore neurovirulence comparable to that of Sabin 2 with the single A481G reversion alone.

Six nOPV2 isolates were found to be recombinants between nOPV2 and Sabin 1 or a species C EV, resulting in the loss of nOPV2 3D polymerase mutations, although no substantial effect on attenuation is expected. However, we detected a double recombinant strain in a sewage sample collected from the Kisenyi ES site in Kabarole on 15 February 2022. This strain had replaced genomic regions upstream and downstream of the capsid coding region with sequences from EV C strains, resulting in the loss of all genetic modifications contributing to nOPV2's genetic stability and attenuation. This double recombinant nOPV2 strain was the first instance of such an event, reported in the May 2022 report of the nOPV Working Group (nOPV WG). It triggered a 'level 2' alert, which prompted a targeted response, including further investigation, communication with stakeholders and the development of action plans to monitor and address subsequent findings. Further laboratory testing confirmed the initial results, and a total of five double recombinant isolates with nearly identical whole-genome sequences were identified from the same sewage sample.

Extensive surveillance confirmed that these isolates remained the only double recombinant nOPV2 viruses detected in Uganda. The absence of related viruses in samples collected before and after 15 February 2022, or from any other site in Uganda, suggests that the virus did not persist or spread widely. Furthermore, the limited VP1 nucleotide changes (2–3) in these isolates indicate minimal replication or transmission before detection. Overall, we confirmed that 7 of 84 children sampled were vaccinated with nOPV2, but given household and community transmission dynamics, some nOPV2 isolates in stool probably originated from secondary spread. The maximum interval between vaccination and a positive stool sample was 46 days, supporting the absence of prolonged shedding, although virus was detected in sewage for up to 109 days after nOPV2 vaccination campaigns. Importantly, sequence analysis provided no evidence of sustained transmission. The absence of any viruses exhibiting >5 VP1 nucleotide changes from the nOPV2 vaccine strain provides additional assurance that no significant transmission events occurred following the nOPV2 vaccination campaigns in Uganda.

High-resolution whole-genome sequence analysis revealed minor local transmission events lasting up to 46 days, but these can probably be explained by continued excretion of the virus by one or a few individuals.

The loss of nOPV2 genetic modifications through double recombination is expected to restore the virus's neurovirulence, increasing the risk of paralytic disease and potential cVDPV2 outbreaks. Indeed, the double recombinant nOPV2 strain from Uganda exhibited high neurovirulence in transgenic mice expressing the human poliovirus receptor, comparable to the MEF-1 wild PV type 2 strain. However, Uganda's high-quality nOPV2 vaccination campaigns and historically generally high population immunity probably prevented spread of this or other

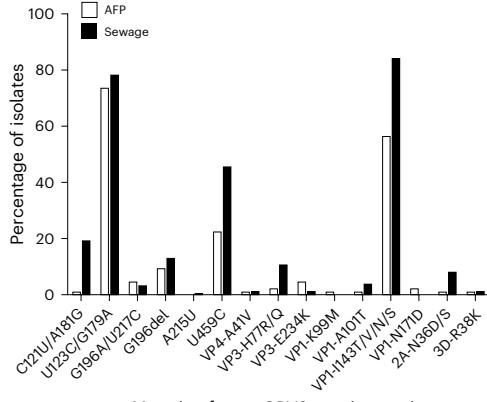

**Fig. 3 | Percentage of nOPV2 isolates with key mutations from the nOPV2 vaccine strain.** The nOPV2 isolates were from AFP (white bars) and ES (black bars) surveillance in Uganda. Source data are provided.

## Table 2 | Location and possible effect of key mutations found in nOPV2 isolates from Uganda

| Nucleotide change | Location in genome | Possible effect |
|---|---|---|
| C121U/A181G | cre5 | Extends stem by 1 bp (121/181) |
| U123C/G179A | cre5 | Strengthens base pair |
| G196A/U217C | Stem-loop II, 5′NTR | Strengthens base pair |
| G196del | Stem-loop II, 5′NTR | Strengthens base pair at the end of a stem and a deletion in a loop |
| A215U | Stem-loop II, 5′NTR | Creates base pair |
| U459C | Stem-loop IV, 5′NTR | Strengthens base pair |
| VP4-A41V | VP4 | Vaccine single nucleotide polymorphism; increases capsid thermostability, marginally increases attenuation |
| VP3-H77R/Q | VP3 | Considered by many to be part of antigenic site 3b |
| VP3-E234K | VP3 | Vaccine single nucleotide polymorphism; external, no predicted contacts; no effect on attenuation |
| VP1-K99M | VP1 | Considered by many to be part of antigenic site 1 |
| VP1-A101T | VP1 | Considered by many to be part of antigenic site 1 |
| VP1-I143T/V/N/S | VP1 | Reversion of secondary attenuation determinant |
| VP1-N171D | VP1 | Alone, slightly decreases attenuation; together with VP1-E295K, no significant impact on attenuation. |
| 2A-N36D/S | 2A | Not a known 2A Vero adaptation; observed in 4 independent isolates |
| 3D-R38K | 3D | Reversion of REC-1 modification; increase in recombination rate and genetic diversity of quasispecies expected |

## Table 3 | Neurovirulence of the double recombinant nOPV2 strain from Kabarole in transgenic mice compared with type 2 vaccine, revertant and wild-type poliovirus strains

| Virus strain | PD$_{50}$ (log$_{10}$) |
|---|---|
| nOPV2 (vaccine strain) | >8.4 |
| Sabin 2 (vaccine strain) | 5.8 (5.1–6.4) |
| S2/481G (Sabin 2 revertant strain) | 1.8 (1.4–2.2) |
| ENV-UGA-KAB-KAB-KIS-22-004 (Kabarole double recombinant strain) | 1.1 (0.6–1.6) |
| MEF-1 (wild-type 2 strain) | 0.7 (0.2–1.2) |

Transgenic mice expressing the poliovirus receptor (minimum $n=8$ biologically independent mice per dilution) were inoculated intraspinally with stocks of representative poliovirus strains. The PD$_{50}$ was calculated using the Spearman–Karber method. Data are presented as PD$_{50}$ values with 95% CI.

5′NTR-relocated CRE and acquire a functional wild-type CRE from an EV-C strain. In contrast, both Sabin 2 single and double recombinant viruses are frequent and exhibit recombination across multiple sites in the P2 and P3 coding regions[20–22].

These findings highlight the essential role of a functional CRE motif in poliovirus replication and suggest that nOPV2 strains with two functional CRE sites are not evolutionarily favoured. While nOPV2 strains can regain full virulence through double recombination with EV-C strains, as demonstrated here, the likelihood of such events and the biological factors driving the generation and selection of double recombinants in humans remain poorly understood. The high prevalence and genetic diversity of EV-C strains capable of recombining with poliovirus, as identified in sewage samples from the Kisenyi ES site in Kabarole, indicate that recombination events are not uncommon. Variations in the prevalence of such EV-C strains across different regions and seasons may influence the evolutionary trajectory of nOPV2 in natural settings. These insights could inform immunization strategies to minimize the risk of generating recombinant viruses.

Our study has several limitations. First, the adequacy of surveillance in areas of use may have influenced the analysis, as the source data rely on the sensitivity of both AFP and ES systems. Second, the presence of multiple nOPV2 isolates in several environmental surveillance samples could have impacted the accuracy of frequency and temporal analyses. Therefore, ongoing monitoring of isolates from diverse field settings and locations will be essential to validate or refine the observations presented here. In addition, integrating these findings with clinical case characteristics, safety data and other epidemiologic factors will be critical for assessing the impact on disease or outbreak dynamics.

The detection of double recombinant nOPV2 strains highlights the need for continued vigilant surveillance and rapid detection of such incidents, as well as intensified vaccination efforts, including emphasis on inactivated polio vaccine use, to accelerate progress towards polio eradication.

## Methods

### Sampling, virus isolation, ITD and shipping of poliovirus materials

AFP surveillance, well established in Uganda, was enhanced to ensure the effective detection of any potential polio paralytic cases. ES, operational in Uganda since May 2017, was expanded to include 11 sites sampled once a month. Following the cVDPV2 outbreak in the country, sampling frequency for Kampala sites was increased to twice a month to enhance sensitivity for PV detection. Stool samples were collected through the routine nationwide AFP surveillance programme, in which children <15 years old presenting to health facilities with weakness or paralysis are investigated regardless of vaccination status. Two stool samples, collected 24–48 h apart, were transported under cold chain to the laboratory for analysis. Grab sewage samples were collected at

nOPV2 strains. In contrast, following this event, similar double recombinant strains have been associated with cVDPV2 outbreaks in countries with less-robust public health systems[15–17]. According to WHO/UNICEF Estimates of National Immunization Coverage, Uganda has consistently maintained coverage above 80% for both OPV3 and inactivated polio vaccine over the years. In contrast, countries such as Nigeria, where transmission of double recombinant nOPV2 polioviruses has been reported, have had estimates for OPV3 and inactivated polio vaccine consistently below 70% (ref. 18; https://immunizationdata.who.int/dashboard/regions/african-region/UGA).

Although nOPV2 has been linked to rare cVDPV2 outbreaks, a recent review of its use up to 1 August 2024 concluded that it has a substantially lower rate of reversion compared with mOPV2. On the basis of mOPV2 and nOPV2 use in Africa in recent years, the number of VDPV2 emergences derived from nOPV2 so far is estimated to be 76% lower than what would have occurred had mOPV2 been used instead[19].

The role of recombination in the evolution of nOPV2 and its implications for viral attenuation deserve further investigation. Whole-genome sequencing of nOPV2 isolates following human replication, as reported here and elsewhere, reveals that genetic modifications in nOPV2, particularly the relocation of the CRE motif to the 5′NTR, appear to limit the frequency and diversity of recombinant structures in comparison to Sabin 2-derived viruses. So far, all identified single recombinant nOPV2 strains retain both the 5′NTR CRE and the KO CRE in the 2C genomic region, while double recombinant viruses lose the

regular intervals from the 11 established ES sites and similarly transported. These collections occurred after nationwide nOPV2 campaigns targeting all children <5 years old.

Stool samples were processed for poliovirus isolation following the standard WHO protocol[23]. Poliovirus isolation was performed using two WHO-recommended cell lines: L20B and RD. L20B cells, which are genetically engineered mouse L cells expressing the human poliovirus receptor CD155, provide a highly specific substrate for poliovirus replication and facilitate rapid differentiation from non-polio enteroviruses. RD cells (human rhabdomyosarcoma cell line), which support the growth of a broader range of enteroviruses, were used in parallel to ensure the sensitive detection of poliovirus. The cells were obtained from master cell banks stored at MHRA which are distributed through the World Health Organization's Global Polio Laboratory Network (GPLN) and are used routinely in accredited national and regional polio laboratories. Sewage samples were collected using the grab method and concentrated overnight using the two-phase separation method[24], followed by poliovirus isolation using the cell culture algorithm following WHO guidelines for ES samples[25]. All cell culture flasks with positive cytopathic effect (CPE) in the L20B cell line were subjected to ITD PCR[26] as independent specimens. The nOPV2 isolates were then inactivated by spotting the supernatant onto FTA cards (Whatman, Life Sciences), which contain chemicals that inactivate the virus while preserving the viral RNA. This method has been used in the WHO Global Polio Laboratory Network for several years and includes a '70 °C for 4 min' heat treatment before spotting isolates on the FTA cards and shipment as non-infectious materials. The FTA cards were sent to the MHRA WHO Global Specialized Laboratory for Polio in the United Kingdom for whole genome sequence analysis. In addition, PV2+/nOPV2− virus isolates were shipped to MHRA for whole-genome sequencing. Due to the trigger of a level 2 response in relation to nOPV2 use, the original sewage concentrate plus previous and subsequent sewage concentrates from the same location ($n = 5$) were also shipped to MHRA for subsequent re-culture using WHO guidelines for ES samples and characterization of any polioviruses.

### PV whole-genome and tiled PCR with reverse transcription amplification

Using disposable punches, 6 discs of 4 mm diameter were cut from each FTA card and viral RNA extracted using the Qiagen QIAamp Viral RNA kit (Qiagen, 52904). Viral RNA was purified from infected cell culture supernatant using the High Pure viral RNA kit (Roche, 11858882001) with Proteinase K pre-treatment. Whole-genome PV PCR with reverse transcription (RT–PCR) fragments were amplified from extracted viral RNA by one-step RT–PCR using a SuperScript III One-Step RT–PCR system with Platinum *Taq* High Fidelity DNA Polymerase (Invitrogen, 12574026) and primers PCR-F (5′-AGA GGC CCA CGT GGC GGC TAG-3′) and PCR-3′ (5′-CCG AAT TAA AGA AAA ATT TAC CCC TAC A-3′)[27]. Amplification conditions were 50 °C for 30 min for the RT reaction plus 94 °C for 2 min plus 42 cycles of 94 °C for 15 s, 55 °C for 30 s and 68 °C for 8 min, with a final extension step of 68 °C for 5 min. Since the quality of RNA extracted from FTA cards is generally poor compared with RNA extracted from cell culture fluid, a custom-made tiled PCR approach with overlap between amplicons was used to generate sequence information spanning the whole genome of any poliovirus present in the FTA card. We followed a published protocol[28]. Primers used to generate tiled PCR products are provided in Extended Data Table 1. Equimolar mixtures of the overlapping PCR products spanning the whole genome were used for sequencing, and additional PCRs were conducted to complete any sequencing gaps if required. Good laboratory practice was used in all molecular assays to prevent cross-contamination of samples. Positive and negative RNA extraction and PCR controls were included in every assay.

### PV whole-genome next-generation sequencing

PV whole-genome and tiled PCR products were sequenced using MiSeq Illumina technology. Sequencing libraries were constructed using the Nextera DNA Prep kit (Illumina, 20060059) and dual-indexed using Nextera DNA Indexes (Illumina, set A 20091660). These libraries were pooled in equimolar concentrations and sequenced with 250-bp paired-end reads on MiSeq v2 (500 cycles) kits (Illumina, MS-102-2003) or with 150-bp paired-end reads on NextSeq 2000 (Illumina, MS-20100984). Initial demultiplexing was performed on board using bcl2fastq v.2.2 (MiSeq) and DRAGEN BCL Convert v.3.8.4 (NextSeq 2000) software. FASTQ sequencing data were adapter and quality trimmed using Cutadapt v.2.10 for a minimum Phred score of Q30, minimal read length of 75 bp and 0 ambiguous nucleotides.

In addition, selected PV whole-genome and tiled PCR products were also sequenced using Oxford Nanopore Technologies (ONT). PCR products underwent a standard preparation for Nanopore sequencing with the SQK-NBD110-9/SQK-NBD112.96 kit together with the barcoding expansion EXP-PBC096 kit. Briefly, PCR products were sheared using the NEBNext Ultra II End Repair/dA-Tailing Module (E7546L), and the native barcode adapters were ligated to the PCR product using Blunt/TA Ligase Master Mix (M0367). Individual barcoded samples were pooled into one tube and the barcoded libraries were purified using AmPure XP beads (A63882). Next, the sequencing adapters supplied in the kit were ligated to the pooled library using NEBNext Quick Ligation Module (E6056L). The flow cells were primed using flow cell priming kit (EXP-FLP002) and the DNA libraries were loaded along with sequencing buffer and loading beads. Sequencing was performed on an ONT MinION Mk1B sequencer using R9.4, R10.3 or R10.4 flow cells. Nanopore data were drawn from several individual MinION sequencing runs. Sequencing run duration ranged between 4 h and 16 h depending on the number of samples included in that run. Only flow cells with a minimum of 600 active pores on flow cell check (pre-test) were used. Fast5 files were base called with the high-accuracy basecalling model using Guppy v.5 with graphics processing unit (GPU) acceleration on a Precision 7540 Dell laptop running Ubuntu 18 LTS with 32 GB RAM, a 16 core Intel Xeon central processing unit (CPU) and Nvidia RTX3000 GPU. FASTQ files were generated using MinKNOW software and stored for further processing.

Fastq files from both Nanopore and Illumina sequencing were further processed and analysed using Geneious 10.2.6 software. Raw sequence data were imported into Geneious 10.2.6 and sequence contigs were built by reference-guided assembly as previously validated[27,29–32]. Fastq reads were initially mapped simultaneously to Sabin 1, nOPV2 and Sabin 3 whole-genome sequences, and contig sequences were generated. Potential nOPV2 recombinant sequences were identified in contigs showing partial sequence coverage across the genome. Filtered reads were then iteratively reassembled to consensus sequences covering the nOPV2 capsid region using stringent assembly conditions to build whole-genome contig sequences. Assembly conditions were as follows: minimum of 50 base overlap, minimum overlap identity of 98%, maximum of 2% mismatches per read and both end-pair reads mapping for Illumina and minimum of 2,000 base overlap, minimum overlap identity of 95%, maximum of 5% mismatches per read for Nanopore. Final consensus sequences were obtained by assigning the most common nucleotide sequence to each nucleotide position within each contig, requiring a minimum of 20-nt sequence coverage, trimming to the reference sequence and removing the primer sequences from the final consensus sequence. When necessary and to confirm the results, fastq reads were also independently assembled de novo, using the same stringent assembly conditions. In addition, the options to produce scaffolds and ignore words repeated more than 100–1,000 times, available in the Geneious assembler, were selected to improve the quality of the assembly process. When sequencing whole-genome PCR products with Nanopore, assembly and contig generation was much more straightforward as it simply required a single step mapping whole-genome reads simultaneously to Sabin 1, nOPV2 and Sabin 3 whole-genome sequences, and generating consensus sequences

as above. Results using different PCR amplification strategies and assembly approaches were identical. Manual analyses for visualizing and quantifying assembly results were performed throughout the process. Final consensus sequences are available from GenBank under accession numbers listed in Supplementary Table 1. nOPV2 genome sequences were aligned using the programme MUSCLE (v.3.8.425), and phylogenetic trees were constructed using the FastTree algorithm (v.2.1.11), both within Geneious.

### Transfection of RNA extracted from FTA cards

Transfection experiments were carried out to show viability of the recombinant virus genetic information obtained from the FTA cards by sequencing on the Illumina platform. Fresh RNA was extracted as described above on the day of transfection. Transfection was carried out using Hank's balanced salt solution and the DEAE-dextran transfection method[33]. The flasks with RNA and the control were incubated at 35 °C in a $CO_2$ incubator. CPE (4+) was observed for flasks transfected with RNA extracted from FTA cards ($n = 3$) on day 3. No CPE was observed in the control flasks at 5 days post transfection. Flasks showing CPE were then subjected to RNA extraction and whole-genome pan-PV PCR amplification, and sequenced using Illumina and Nanopore technology as described above[27].

### Direct sequencing of enteroviruses from sewage concentrates

A retrospective investigation was conducted to determine the EV distribution in sewage concentrates collected from January 2022 to April 2022 from the Kisenyi site, that is, before and after the point detection of the double recombinants ($n = 5$). RNA was extracted from the sewage concentrates using the High Pure viral RNA kit (Roche, 11858882001) with Proteinase K pre-treatment, and pan-EV entire capsid-coding region RT–PCR templates were generated and sequenced using the MiSeq Illumina platform as described here and elsewhere[32]. The closest virus relatives to the Uganda EV sewage strains were identified using the RIVM and BLAST online sequence analysis tools[34,35], and EV serotypes were assigned on the basis of their VP1 sequence. EV genome sequences were aligned using the programme MUSCLE (v.3.8.425), and phylogenetic trees were constructed using the FastTree algorithm (v.2.1.11), both within Geneious.

### Transgenic mouse neurovirulence test

Tg66-CBA transgenic mice expressing the human poliovirus receptor (50% male, 50% female, 6–8 weeks old) were used for these experiments. Tg66-CBA mice are the product of crossing Tg66 mice with CBA/J mice several times then selecting for homozygous CBA MHC genes and homozygous PVR. Animal experiments were performed at the MHRA, with ethics approval from MHRA's Ethics and Human Materials Advisory Committees and the Animal Welfare and Ethical Review Body. All procedures were conducted under UK Home Office Procedure Project Licence Number PPL PP6108158. The mice were housed in individually ventilated cages or conventional cages under controlled environmental conditions in accordance with the UK Animals (Scientific Procedures) Act 1986 and the Home Office Code of Practice for the Housing and Care of Animals Bred, Supplied or Used for Scientific Purposes[36]. Ambient temperature was maintained between 20–24 °C with relative humidity maintained between 45–65%. A daily 12 h/12 h light/dark cycle with half an hour of half-light to mimic dawn and dusk was provided to regulate circadian rhythms. Stocking density was carefully considered to ensure that mice were provided with sufficient floor space and to allow for provision of environmental enrichment in line with legislative standards. Environmental enrichment, including nesting material, refuges, wooden and disposable enrichment, was provided during weekly cage cleaning to encourage natural behaviours that are crucial for maintaining the health and wellbeing of the mice. Diet and water were provided ad libitum, and environmental parameters were continuously monitored to meet legislative and welfare requirements. Mice were routinely handled using refined handling techniques, including tunnel handling and cupping, to minimize stress and ensure high standards of animal welfare. The mice were inoculated via the intraspinal route with 5 μl of tenfold serial virus dilutions (a minimum of 8 mice per dilution) and monitored for clinical signs for 14 days (ref. 37). The cell culture infectious dose ($CCID_{50}$) required to paralyze 50% of the mice ($PD_{50}$) was calculated using the Spearman–Karber method[38]. While the transgenic mouse neurovirulence test (TgmNVT) does not reproduce natural infection through the oral route, it has been proven to accurately measure the neurovirulence of PV isolates and hence their potential for causing paralytic disease, showing very good correlation with the results using the gold standard monkey neurovirulence test and is recommended by the World Health Organization.

### Statistical analysis

Simple linear regression and molecular clock-based inference of the nucleotide sequence data were done using GraphPad Prism (v.10.1.2) software. Regression analyses are exploratory/descriptive (mutation vs time), hence no inferential statistics ($p$ values) were the focus. The dose required to cause paralysis in 50% of transgenic animals ($PD_{50}$) was calculated using the Spearman–Karber method using Microsoft Excel software. Poliovirus sequencing data were processed and analysed using Geneious (v.10.2.6).

### Reporting summary

Further information on research design is available in the Nature Portfolio Reporting Summary linked to this article.

## Data availability

All consensus poliovirus genomic sequences generated in this study have been deposited in the NCBI sequence database with GenBank accession numbers listed in Supplementary Table 1. Raw fastq NGS files generated in this study corresponding to double recombinant viruses and enterovirus sequences from sewage have been deposited in the NCBI Sequence Read Archive under Bio project number PRJNA1295797. Type 2 poliovirus isolates used in this study were obtained through routine public health surveillance and were handled in accordance with WHO's Global Action Plan for Poliovirus Containment (GAPIV). In compliance with international containment requirements, most viral materials have been destroyed or securely stored in designated poliovirus-essential facilities. As such, live virus materials are not generally available for distribution. Researchers may request access to any remaining materials, where applicable, subject to institutional approval, national regulations and WHO containment policies. Metadata associated with each sequence (Lab ID, GenBank ID, sample type, collection date and location) are available in Supplementary Table 1. Source data are provided with this paper.

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

## Acknowledgements

We acknowledge WHO AFRO and Global laboratory coordinators, members of the nOPV Genetic Characterization Sub-Group and nOPV Working Group, BioFarma; R. Mate and P. Sinnakandu of the MHRA NGS unit, S. Zipursky from WHO, and F. Kurji from FDK Consulting. Work at MHRA was supported by funds from the National Institute for Health Research (NIHR) Policy Research Programme (NIBSC Regulatory Science Research Unit), MHRA core funding and a grant from the Bill and Melinda Gates Foundation (INV-043859). The views expressed in the publication are those of the authors and not necessarily those of the funders, NHS, the NIHR, the Department of Health, arm's length bodies, or other government departments.

## Author contributions

J.M., M.M., P.T., C.R.B. and A.M. conceived and designed the experiments; P.T., M.M., S.C., A.S., M.-L.J., D.K., J.C., K.O.A., E.B. and H.B. performed the experiments; J.M., P.T., M.M., L.S., D.K., J.C., K.O.A., K.M.H., B.B., E.D., M. Bessaud, A.S.B. and A.M. analysed the data; J.M., P.T., A.S., J.B., K.M.H., E.B., H.B., D.K., M. Bessaud, A.S.B., C.R.B., M. Birungi and A.M. contributed materials/analysis tools.

## Competing interests

All authors declare no competing interests.

## Inclusion and ethics statement

This research was conducted through an inclusive and collaborative partnership involving institutions across Uganda, the United Kingdom, Israel, France, the Netherlands, the United States and the World Health Organization. The study reflects a shared commitment to global public health, scientific integrity and equitable collaboration. Local expertise at the Uganda Virus Research Institute and the EPI Laboratory team played a central role in sample collection, processing and virus isolation, ensuring contextual relevance and contributing to capacity strengthening.

The study did not involve human participants, but the analysis of viral nucleotide sequences involved viral isolates recovered from anonymous clinical (stools) and environmental (sewage) samples. Clinical and environmental specimens analysed in this study were collected as part of routine public health surveillance under the GPEI. AFP surveillance and environmental sampling are core components of Uganda's National Poliovirus Surveillance programme, implemented by the Ministry of Health with technical support from the World Health Organization. These surveillance activities follow WHO-recommended protocols and are classified as non-research public health activities that do not require individual informed consent. The Uganda Virus Research Institute, serving as the National Polio Laboratory, conducted specimen processing and virus isolation under standard WHO procedures. Only de-identified poliovirus isolates were transferred for sequencing and genomic analysis. No personally identifiable information was collected or analysed; the only human-derived metadata retained were sample collection dates, used solely for epidemiological context. The study did not involve human participants research as defined by international ethics guidelines, and all analyses were limited to viral genomic sequences. Ethics oversight was provided by national authorities, and all activities complied with relevant ethics standards for the secondary use of surveillance-derived human specimens in global public health research.

Animal work was approved by MHRA's Ethics and Human Materials Advisory Committees. MHRA's Animal Welfare and Ethical Review Body approved the application for Procedure Project Licence Number PPL PP6108158, which was approved by the UK Government Home Office and under which animal care and protocols shown in this paper were conducted. All animal care and protocols used at MHRA adhere to the UK Animals (Scientific Procedures) Act 1986 and the Home Office Code of Practice for the Housing and Care of Animals Bred, Supplied or Used for Scientific Purposes.

All authors contributed to the research and writing process with mutual respect for diverse expertise, fostering transparency, inclusivity and shared ownership. The study aligns with principles of responsible data use, equitable access to research opportunities and the global effort to eliminate poliovirus transmission through open, cross-border scientific cooperation.

## Additional information

**Extended data** is available for this paper at https://doi.org/10.1038/s41564-025-02219-w.

**Correspondence and requests for materials** should be addressed to Javier Martin.

**Phionah Tushabe** [1,8]**, Manasi Majumdar**[2,8]**, Sarah Carlyle**[2]**, Lester Shulman**[3]**, Alfred Ssekagiri**[1]**, Marie-Line Joffret**[4]**, Dimitra Klapsa**[2]**, Jeroen Cremer**[5]**, Kafayat O. Arowolo** [2]**, Erika Bujaki**[2]**, Henry Bukenya**[1]**, EPI Laboratory team\*, Mary Bridget Nanteza**[1]**, Irene Turyahabwe**[1]**, Prossy Namuwulya**[1]**, Molly Birungi**[1]**, James Peter Eliku**[1]**, Francis Aine**[1]**, Mayi Tibanagwa**[1]**, Lucy Nakabazzi**[1]**, Joseph Gaizi**[1]**, Arnold Mugagga Ssebuuma**[1]**, Rajab Dhatemwa**[1]**, Charles Okia**[1]**, Mary Nyachwo**[1]**, Mary Bridget Nanteza**[1]**, Irene Turyahabwe**[1]**, Prossy Namuwulya**[1]**, Molly Birungi**[1]**, James Peter Eliku**[1]**, Francis Aine**[1]**, Mayi Tibanagwa**[1]**, Lucy Nakabazzi**[1]**, Joseph Gaizi**[1]**, Arnold Mugagga Ssebuuma**[1]**, Rajab Dhatemwa**[1]**, Charles Okia**[1]**, Mary Nyachwo**[1]**, Kaija M. Hawes**[6]**, Barnabas Bakamutumaho**[1]**, Erwin Duizer**[5]**, Mael Bessaud**[4]**, Ananda S. Bandyopadhyay**[6]**, Andrew Macadam**[2]**, Charles R. Byabamazima**[7]**, Javier Martin** [2,8] ✉ **& Josephine Bwogi** [1,8]

[1]Uganda Virus Research Institute (UVRI), Entebbe, Uganda. [2]Medicines and Healthcare products Regulatory Agency (MHRA), Hertfordshire, UK. [3]Sackler Faculty of Medicine, Tel Aviv University, Tel Aviv, Israel. [4]Institut Pasteur, Paris, France. [5]National Institute for Public Health and the Environment (RIVM), Bilthoven, the Netherlands. [6]Bill and Melinda Gates Foundation, Seattle, WA, USA. [7]World Health Organization Regional Office for Africa (AFRO), Harare, Zimbabwe. [8]These authors contributed equally: Phionah Tushabe, Manasi Majumdar, Javier Martin, Josephine Bwogi. \*A list of authors and their affiliations appears at the end of the paper. ✉e-mail: Javier.Martin@mhra.gov.uk

**EPI Laboratory team**

**Mary Bridget Nanteza**[1]**, Irene Turyahabwe**[1]**, Prossy Namuwulya**[1]**, Molly Birungi**[1]**, James Peter Eliku**[1]**, Francis Aine**[1]**, Mayi Tibanagwa**[1]**, Lucy Nakabazzi**[1]**, Joseph Gaizi**[1]**, Arnold Mugagga Ssebuuma**[1]**, Rajab Dhatemwa**[1]**, Charles Okia**[1] **& Mary Nyachwo**[1]

**Extended Data Table 1 | Sequences and genome location of primers used for PCR amplifications**

| Reaction/ Tile | Forward Primer Name | Forward Primer Sequence (5'-3') | Reverse Primer Name | Reverse Primer Sequence (5'-3') | Amplicon (start-end) | Amplicon length |
|---|---|---|---|---|---|---|
| **Main tiled reactions (MHRA)** | | | | | | |
| 1 | PanPV-WG-Fwd | CCAGAGGCCCACGTGGCGGCTAG | PV-1999-Rev | CAAGTTGAGGGGTATCATGGTGT | 29-1999 | 1971 |
| 2 | PanSabin-Seg2-Fwd* | TCTGCCCRGTKGATTAYCTC | Q8-Rev | AAGAGGGTCTCTRTTCCACAT | 1533-3565 | 2033 |
| 3 | PanSabin-Seg3-Fwd* | GTMAATGATCACAACCC | PanSabin-Seg3-Rev* | GTTGGAAAGTTGTACATTAG | 3275-5922 | 2648 |
| 4 | PV-5455-Fwd | GGRCAYCARGGWGCATAYACTG | PanPV-WG-Rev | CCGAATTAAAGAAAAATTTACCCCTACA | 5423-7498 | 2077 |
| **Additional reactions (MHRA)** | | | | | | |
| Reaction/ Tile | Forward Primer Name | Forward Primer Sequence (5'-3') | Reverse Primer Name | Reverse Primer Sequence (5'-3') | Amplicon (start-end) | Amplicon length |
| 5 | nOPV2-1864-Fwd | AACACTCCAGGGAGTAACCAGTA | Q8-Rev | AAGAGGTCTCRTTCACAT | 1844-3564 | 1721 |
| 6 | Y7R-Fwd | GGTTTTGTGTCAGCITGYAAYGA | nOPV-cre-Rev | TCGATACGGTGCTTGGATTTAAATTG | 2462-4545 | 2083 |
| 7 | Y7R-Fwd | GGTTTTGTGTCAGCITGYAAYGA | Ari-Rev | TCAATACGGTGTTTGCTCTTGAACTG | 2462-4545 | 2083 |
| 8 | PanSabin-Seg3-Fwd* | GTMAATGATCACACACC | PV-5643-Rev | CWCCWGGTGARGCRTGRGTTGG | 3275-5630 | 2356 |
| 9 | PV-4440-Fwd | CAGTCCAAGAGGTTTGCACC | PanPV-WG-Rev | CCGAATTAAAGAAAAATTTACCCCTACA | 4439-7498 | 3060 |
| 10 | nOPV2-5439-Fwd | CTGGGCACCAGGGTGCATA | PanPV-WG-Rev | CCGAATTAAAGAAAAATTTACCCCTACA | 5421-7498 | 2078 |
| 11 | Y7R-Fwd | GGTTTTGTGTCAGCITGYAAYGA | PanPV-WG-Rev | CCGAATTAAAGAAAAATTTACCCCTACA | 2462-7498 | 5036 |
| 12 | PanPV-WG-Fwd | CCAGAGGCCCACGTGGCGGCTAG | Q8-Rev | AAGAGGGTCTCRTTCACAT | 29-3564 | 3536 |
| 13 | PanSabin-Seg3-Fwd* | GTMAATGATCACACACC | PanPV-WG-Rev | CCGAATTAAAGAAAAATTTACCCCTACA | 3275-7498 | 4224 |
| 14 | nOPV2-2845-Fwd | CAGCAGATTGTTTTCGGTT | PanPV-WG-Rev | CCGAATTAAAGAAAAATTTACCCCTACA | 2845-7498 | 4653 |
| **Additional reactions (IPP)** | | | | | | |
| Reaction/ Tile | Forward Primer Name | Forward Primer Sequence (5'-3') | Reverse Primer Name | Reverse Primer Sequence (5'-3') | Amplicon (start-end) | Amplicon length |
| 15 | Sab2-1472-Fwd | TCTGCCCRGTKGATTAYCTC | nOPV-4545-Rev | TCGATACGGTGCTTGGATTTAAATTG | 1533-4545 | 3013 |
| 16 | Sab2-4165-Fwd | GCAAAGGGACTGGAGTGGG | PanPV-WG-Rev | CCGAATTAAAGAAAAATTTACCCCTACA | 4226-7498 | 3273 |

Numbering as in nOPV2 vaccine strain (GenBank ID: MZ245455) IUPAC codes: R = A/G, Y = C/T, K = G/T, W = A/T, M = A/C *From Valesano AL, Taniuchi M, Fitzsimmons WJ, Islam MO, Ahmed T, Zaman K, Haque R, Wong W, Famulare M, Lauring AS. The Early Evolution of Oral Poliovirus Vaccine Is Shaped by Strong Positive Selection and Tight Transmission Bottlenecks. Cell Host Microbe. 2021 Jan 13;29(1):32-43.e4. doi: 10.1016/j.chom.2020.10.011. Epub 2020 Nov 18. PMID: 33212020; PMCID: PMC7815045.

**Extended Data Table 2 | Number of nOPV2 isolates with different combinations of mutations reducing attenuation of the virus**

| Total_mutations | No. of isolates | C121T | T123C | G147A | G179A | A181G | Deletion_179_183 | G196A | G196del | A215T | T217C | G322A | T459C | VP1-I143T/V/N/S | VP1-N171D |
|---|---|---|---|---|---|---|---|---|---|---|---|---|---|---|---|
| 5 | 1 | | ✓ | | | ✓ | | ✓ | | | | | ✓ | ✓ | |
| 4 | 13 | | | | ✓ | ✓ | | | | | | | ✓ | ✓ | |
| 4 | 6 | | ✓ | | | | | | ✓ | | | | ✓ | ✓ | |
| 4 | 4 | | ✓ | | | ✓ | | | | | | | ✓ | ✓ | |
| 4 | 3 | | ✓ | | | | | | | | ✓ | | ✓ | ✓ | |
| 4 | 1 | | | | ✓ | | | | | | | ✓ | ✓ | ✓ | |
| 4 | 1 | | | | ✓ | ✓ | | | | | | ✓ | | ✓ | |
| 4 | 1 | | ✓ | | | | | | | ✓ | | | ✓ | ✓ | |
| 4 | 1 | | ✓ | ✓ | | | | | | | | | ✓ | ✓ | |
| 4 | 1 | ✓ | | | | ✓ | | | | | | | ✓ | ✓ | |
| 3 | 37 | | ✓ | | | | | | | | | | ✓ | ✓ | |
| 3 | 8 | | ✓ | | | | | | ✓ | | | | | ✓ | |
| 3 | 7 | | | | ✓ | | | | | | | | ✓ | ✓ | |
| 3 | 5 | | | | ✓ | ✓ | | | | | | | | ✓ | |
| 3 | 2 | | | | | ✓ | | | | | | | ✓ | ✓ | |
| 3 | 2 | | | | ✓ | | | | ✓ | | | | | ✓ | |
| 3 | 2 | | ✓ | | | | | ✓ | | | | | | ✓ | |
| 3 | 2 | | ✓ | | | ✓ | | | | | | | | ✓ | |
| 3 | 1 | | | | | | ✓ | | | | | | ✓ | ✓ | |
| 3 | 1 | | ✓ | | | | | | | | | ✓ | | ✓ | |
| 3 | 1 | | ✓ | | | | | | | | ✓ | | | ✓ | |
| 2 | 37 | | ✓ | | | | | | | | | | | ✓ | |
| 2 | 11 | | | | | | | | ✓ | | | | | ✓ | |
| 2 | 10 | | | | ✓ | | | | | | | | | ✓ | |
| 2 | 4 | | ✓ | | | | | | | | | | ✓ | | |
| 2 | 1 | | | | | | | | | | | | ✓ | ✓ | |
| 2 | 1 | | | | ✓ | | | | | | | | | | ✓ |
| 2 | 1 | | | | ✓ | | | | | | | | ✓ | | |
| 2 | 1 | | | | ✓ | | | | | | ✓ | | | | |
| 1 | 17 | | ✓ | | | | | | | | | | | | |
| 1 | 14 | | | | | | | | | | | | | ✓ | |
| 1 | 7 | | | | ✓ | | | | | | | | | | |
| 1 | 1 | | | | | | | | | | | | | | ✓ |
| 1 | 1 | | | | | | | | | | | | ✓ | | |
| 1 | 1 | | | | | | | | | | ✓ | | | | |
| 0 | 24 | | | | | | | | | | | | | | |
| TOTAL | 231 | 1 | 126 | 1 | 51 | 28 | 1 | 3 | 27 | 1 | 7 | 3 | 87 | 175 | 2 |

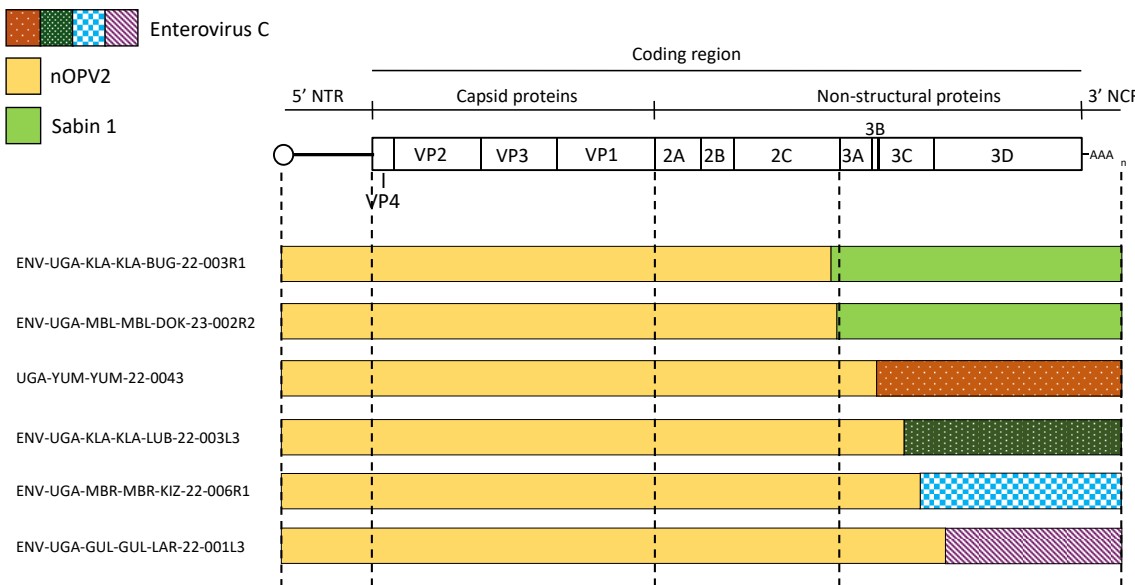

**Extended Data Fig. 1 | Schematic representation of nOPV2 single recombinant strains from Uganda.** Sequences derived from nOPV2, Sabin 1 and unidentified enterovirus C strains are indicted in different colours.

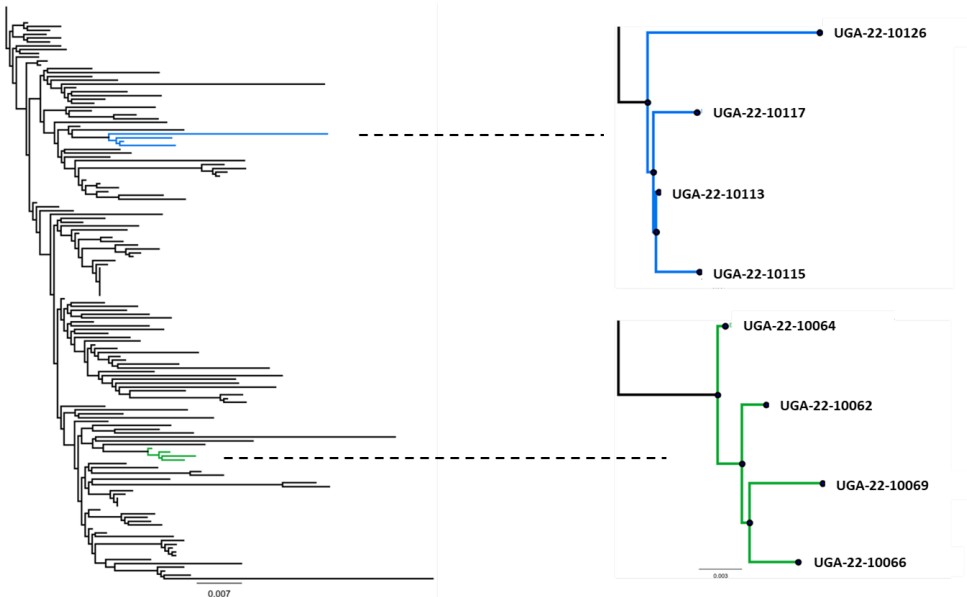

**Extended Data Fig. 2 | Phylogenetic relationships between nOPV2 ES isolates from Uganda.** Phylogenetic trees with all nOPV2 sequences from ES samples were constructed using the FastTree algorithm (version 2.1.11) within Geneious. Sequence alignment was first generated using MUSCLE, and trees were inferred under the generalized time-reversible (GTR) model with CAT approximation for rate heterogeneity across sites. FastTree constructs an initial tree using a neighbor-joining heuristic, followed by maximum-likelihood optimization with nearest-neighbor interchanges (NNIs) to refine topology and branch lengths. Trees were visualized and annotated using Geneious Tree Viewer. The scale bar represents substitutions per site. Two small genetic clusters (in blue and green in the tree) were identified.

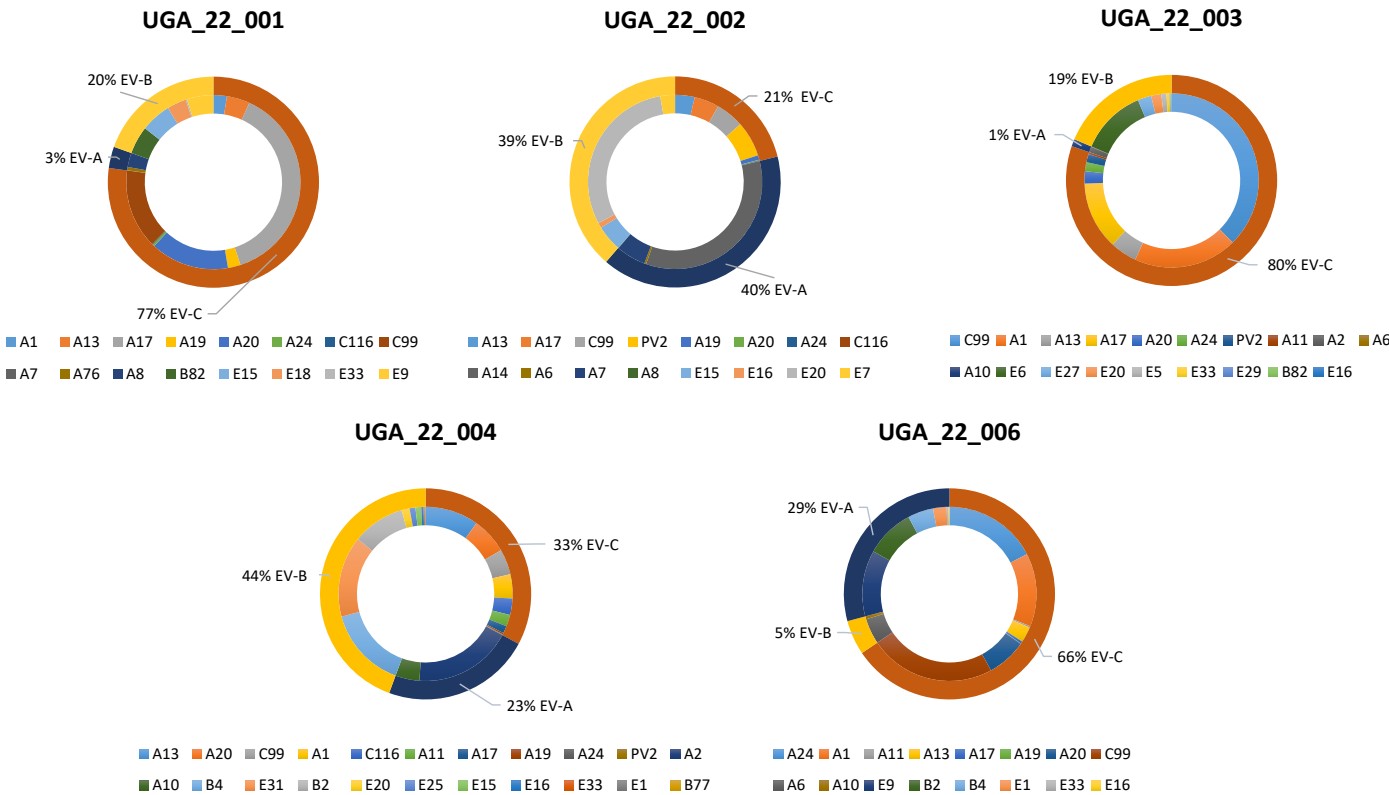

**Extended Data Fig. 3 | Prevalence of species A, B and C enterovirus strains in sewage samples from the Kisenyi Sewage Ponds site in Kabarole (Uganda).** Filtered fastq reads from RT-PCR products from the sewage concentrates were mapped to consensus sequences obtained for each sample. The proportions of reads mapping to each EV serotype in each sample are shown with different colours as indicated in the figure. Source data are provided as a Source Data file.

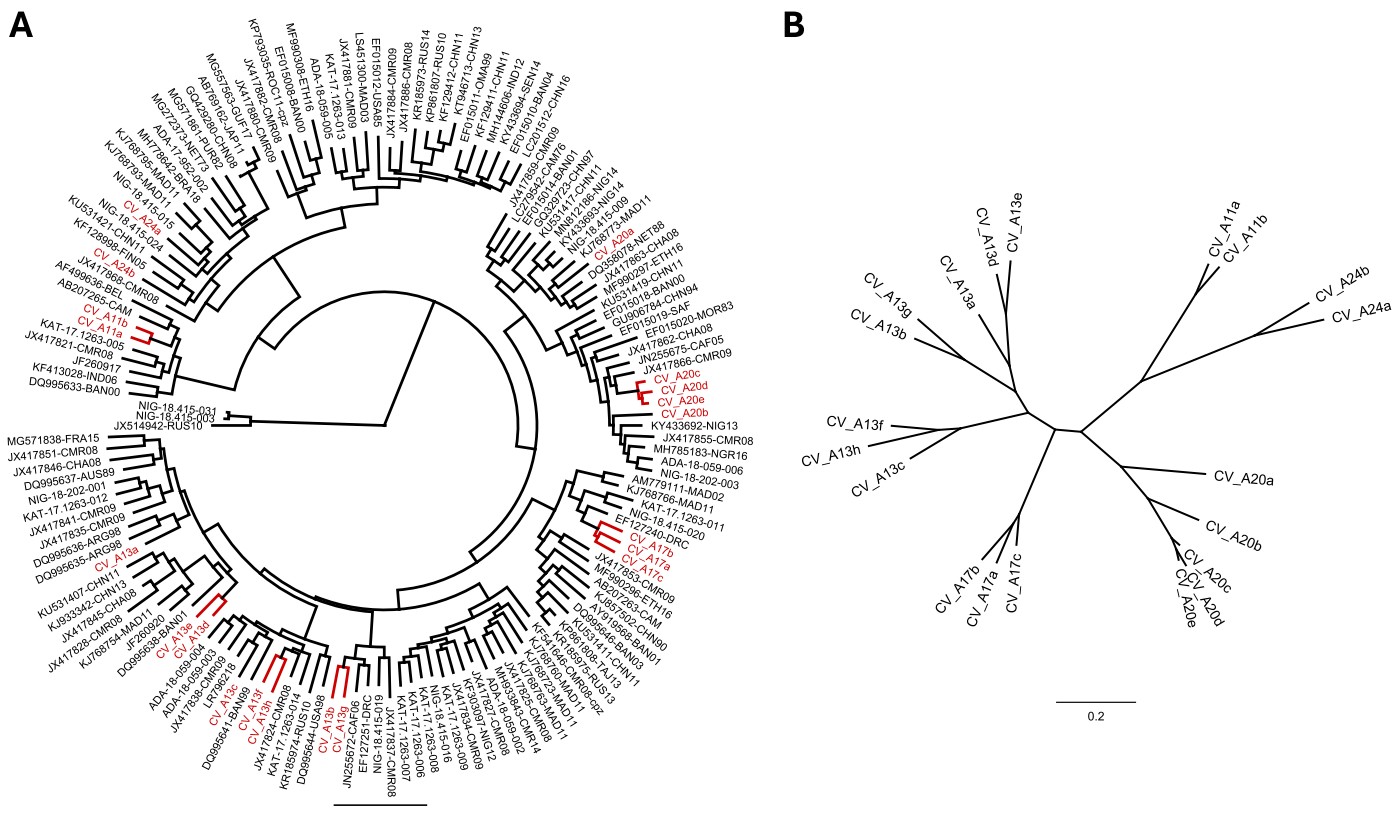

**Extended Data Fig. 4 | Analysis of non-polio enterovirus C VP1 sequences from Uganda.** Phylogenetic relationships between VP1 nucleotide sequences of species C enterovirus strains from serotypes known to frequently recombine with poliovirus identified in sewage samples from the Kisenyi Sewage Ponds site in Kabarole (Uganda). Phylogenetic trees were constructed using the FastTree algorithm (version 2.1.11) within Geneious. Sequence alignment was first generated using MUSCLE, and trees were inferred under the generalized

time-reversible (GTR) model with CAT approximation for rate heterogeneity across sites. FastTree constructs an initial tree using a neighbor-joining heuristic, followed by maximum-likelihood optimization with nearest-neighbor interchanges (NNIs) to refine topology and branch lengths. Trees were visualized and annotated using Geneious Tree Viewer. The scale bar represents substitutions per site. VP1 sequences from Ugandan strains in this study are shown in red text in panel **A** and analysed alone in panel **B**.

# Reporting Summary

## Statistics

For all statistical analyses, confirm that the following items are present in the figure legend, table legend, main text, or Methods section.

| n/a | Confirmed | |
|---|---|---|
| ☐ | ☒ | The exact sample size (*n*) for each experimental group/condition, given as a discrete number and unit of measurement |
| ☒ | ☐ | A statement on whether measurements were taken from distinct samples or whether the same sample was measured repeatedly |
| ☐ | ☒ | The statistical test(s) used AND whether they are one- or two-sided<br>*Only common tests should be described solely by name; describe more complex techniques in the Methods section.* |
| ☐ | ☒ | A description of all covariates tested |
| ☒ | ☐ | A description of any assumptions or corrections, such as tests of normality and adjustment for multiple comparisons |
| ☐ | ☒ | A full description of the statistical parameters including central tendency (e.g. means) or other basic estimates (e.g. regression coefficient) AND variation (e.g. standard deviation) or associated estimates of uncertainty (e.g. confidence intervals) |
| ☐ | ☒ | For null hypothesis testing, the test statistic (e.g. *F*, *t*, *r*) with confidence intervals, effect sizes, degrees of freedom and *P* value noted<br>*Give P values as exact values whenever suitable.* |
| ☒ | ☐ | For Bayesian analysis, information on the choice of priors and Markov chain Monte Carlo settings |
| ☒ | ☐ | For hierarchical and complex designs, identification of the appropriate level for tests and full reporting of outcomes |
| ☐ | ☒ | Estimates of effect sizes (e.g. Cohen's *d*, Pearson's *r*), indicating how they were calculated |

*Our web collection on statistics for biologists contains articles on many of the points above.*

## Software and code

Policy information about availability of computer code

| | |
|---|---|
| Data collection | Data for this study was produced by hardware and software associated with Illumina and Oxford Nanopore sequencing platforms. Metadata associated with study samples was provided in Microsoft Excel worksheets. |
| Data analysis | Simple linear regression statistical analysis and molecular clock-based inference of the nucleotide sequence data were done using GraphPad Prism (version 10.1.2) software. The dose required to cause paralysis in 50% of transgenic animals (PD50) was calculated by the Spearman–Karber method using Microsoft Excel software. Poliovirus sequencing data were processed and analysed using Geneious (version 10.2.6). |

For manuscripts utilizing custom algorithms or software that are central to the research but not yet described in published literature, software must be made available to editors and reviewers. We strongly encourage code deposition in a community repository (e.g. GitHub). See the Nature Portfolio guidelines for submitting code & software for further information.

## Data

Policy information about availability of data

All manuscripts must include a data availability statement. This statement should provide the following information, where applicable:

- Accession codes, unique identifiers, or web links for publicly available datasets
- A description of any restrictions on data availability
- For clinical datasets or third party data, please ensure that the statement adheres to our policy

All consensus poliovirus genomic sequences generated in this study have been deposited in the NCBI sequence database with GenBank accession numbers

PV576826-PV577058. Raw fastq NGS files generated in this study corresponding to double recombinant viruses and enterovirus sequences from sewage have been deposited in the NCBI Sequence Read Archive under project code Bio project number: PRJNA1295797. Metadata associated with each sequence (Lab ID, GenBank ID, sample type, collection date and location) are available as Supplementary Table 3. Source data are provided with this paper.

## Research involving human participants, their data, or biological material

Policy information about studies with human participants or human data. See also policy information about sex, gender (identity/presentation), and sexual orientation and race, ethnicity and racism.

| | |
|---|---|
| Reporting on sex and gender | N/A |
| Reporting on race, ethnicity, or other socially relevant groupings | N/A |
| Population characteristics | N/A |
| Recruitment | N/A |
| Ethics oversight | The study did not involve human participants but the analysis of viral nucleotide sequences obtained from viral isolates recovered from anonymous clinical (stools) and environmental samples (sewage). Clinical and environmental specimens analyzed in this study were collected as part of routine public health surveillance under the Global Polio Eradication Initiative (GPEI). Acute flaccid paralysis (AFP) collecting stool samples is a core component of Uganda's national poliovirus surveillance program, implemented by the Ministry of Health with technical support from the World Health Organization (WHO). These surveillance activities follow WHO-recommended protocols and are classified as non-research public health activities that do not require individual informed consent. Only de-identified viral sequencing data were analysed in this study. No personally identifiable information was collected or analysed. |

Note that full information on the approval of the study protocol must also be provided in the manuscript.

# Field-specific reporting

Please select the one below that is the best fit for your research. If you are not sure, read the appropriate sections before making your selection.

☒ Life sciences          ☐ Behavioural & social sciences          ☐ Ecological, evolutionary & environmental sciences

For a reference copy of the document with all sections, see nature.com/documents/nr-reporting-summary-flat.pdf

# Life sciences study design

All studies must disclose on these points even when the disclosure is negative.

| | |
|---|---|
| Sample size | No statistical methods were used to predetermine sample size. The study included all available clinical and environmental specimens collected through routine poliovirus surveillance activities conducted under the Global Polio Eradication Initiative (GPEI) in Uganda over the study period. Sample inclusion was determined by availability and detection of type 2 polioviruses, rather than experimental design, and is representative of field surveillance during and after nOPV2 campaigns. |
| Data exclusions | No data were excluded from analysis unless sequencing quality was insufficient for reliable interpretation (e.g., partial genome coverage or failed sequencing runs). |
| Replication | All virus isolation, molecular testing, and sequencing were performed using standard WHO-accredited protocols. Laboratory processes were carried out in reference laboratories that undergo annual WHO proficiency testing and accreditation. Key findings (e.g., presence of recombinant genomes) were independently verified through repeated sequencing or analysis pipelines where appropriate. No experimental replication was performed beyond standard confirmation procedures due to the descriptive nature of the surveillance data. |
| Randomization | Randomization was not applicable, as the study is based on observational surveillance data and not experimental assignment. Samples were tested in the order received, following standard laboratory workflows. |
| Blinding | Blinding was not applicable. Laboratory personnel conducting virus isolation and sequencing were aware of the sample origin (clinical or environmental), but no analytical decisions were influenced by these factors. Bioinformatic analysis was conducted based on sequence data without reference to the sample identity beyond metadata needed for interpretation. |

# Reporting for specific materials, systems and methods

We require information from authors about some types of materials, experimental systems and methods used in many studies. Here, indicate whether each material, system or method listed is relevant to your study. If you are not sure if a list item applies to your research, read the appropriate section before selecting a response.

## Materials & experimental systems

| n/a | Involved in the study |
|-----|----------------------|
| ☒ | ☐ Antibodies |
| ☐ | ☒ Eukaryotic cell lines |
| ☒ | ☐ Palaeontology and archaeology |
| ☐ | ☒ Animals and other organisms |
| ☒ | ☐ Clinical data |
| ☒ | ☐ Dual use research of concern |
| ☒ | ☐ Plants |

## Methods

| n/a | Involved in the study |
|-----|----------------------|
| ☒ | ☐ ChIP-seq |
| ☒ | ☐ Flow cytometry |
| ☒ | ☐ MRI-based neuroimaging |

# Eukaryotic cell lines

Policy information about cell lines and Sex and Gender in Research

| | |
|---|---|
| Cell line source(s) | Poliovirus isolation was performed using two WHO-recommended eukaryotic cell lines: L20B and RD. L20B cells, which are genetically engineered mouse L cells expressing the human poliovirus receptor CD155, provide a highly specific substrate for poliovirus replication and facilitate rapid differentiation from non-polio enteroviruses. RD cells (human rhabdomyosarcoma cell line), which support the growth of a broader range of enteroviruses, were used in parallel to ensure the sensitive detection of poliovirus. The cells were obtained from master cell banks stored at MHRA which are distributed through the World Health Organization's Global Polio Laboratory Network (GPLN) and are used routinely in accredited national and regional polio laboratories. |
| Authentication | Master cell banks are authenticated using validated molecular typing methods by WHO-designated reference laboratories prior to distribution. Receiving laboratories use these authenticated stocks and operate under WHO quality standards. Additional functional verification of cell lines is performed through annual WHO proficiency testing as part of the laboratory accreditation process. |
| Mycoplasma contamination | All laboratories involved in this study adhere to WHO and Good Laboratory Practice (GLP) standards. Regular quality control measures, including mycoplasma testing, are in place, and no contamination was identified during the study period. |
| Commonly misidentified lines (See ICLAC register) | The RD and L20B cell lines used in this study are not listed in the ICLAC register of commonly misidentified cell lines. Both lines are widely validated and recognized within the Global Polio Eradication Initiative for their specificity and reliability in poliovirus isolation. |

# Animals and other research organisms

Policy information about studies involving animals; ARRIVE guidelines recommended for reporting animal research, and Sex and Gender in Research

| | |
|---|---|
| Laboratory animals | Neurovirulence testing was performed using Tg66-CBA transgenic mice expressing the human poliovirus receptor (PVR), in accordance with WHO recommendations. Mice were 6–8 weeks old (50% male and 50% female). Tg66-CBA mice are the product of crossing Tg66 mice with CBA/J mice several times then selecting for homozygous CBA MHC genes and homozygous PVR. |
| Wild animals | No wild animals were involved in this study. |
| Reporting on sex | Both male and female Tg66 mice were included in approximately equal numbers to account for known sex-related differences in sensitivity to poliovirus neurovirulence. This balance is maintained to ensure robust and interpretable results and is consistent with WHO and ARRIVE recommendations. |
| Field-collected samples | Stool and sewage samples were collected through routine poliovirus surveillance activities under the Global Polio Eradication Initiative (GPEI) in Uganda. All samples were collected, transported, stored, and processed following WHO-recommended protocols and timelines to ensure optimal conditions for the reliable detection of live poliovirus. Environmental samples were obtained from established sewage surveillance sites, and clinical samples were collected through acute flaccid paralysis (AFP) surveillance systems coordinated by the Uganda Ministry of Health. No disturbance to wildlife, natural habitats, or ecosystems occurred during fieldwork. |
| Ethics oversight | Animal experiments were performed at the MHRA, under ethical approval from MHRA's Ethics and Human Materials Advisory Committees and the Animal Welfare and Ethical Review Body (AWERB). All procedures were conducted under UK Home Office Procedure Project Licence Number PPL PP6108158. The mice are housed in Individually Ventilated Cages (IVCs) or conventional cages, under controlled environmental conditions in accordance with the UK Animals (Scientific Procedures) Act 1986 and the Home Office Code of Practice for the Housing and Care of Animals Bred, Supplied or Used for Scientific Purposes (37). Ambient temperature is maintained between 20-24 degrees Celsius with relative humidity maintained between 45-65%. A daily 12:12 hour light/dark cycle with half an hour of half-light to mimic dawn and dusk, is provided to regulate circadian rhythms. Stocking density is carefully considered to ensure the mice are provided with sufficient floor space and to allow for provision of environmental enrichment in line with legislative standards. Environmental enrichment, including nesting material, refuges, wooden and disposable enrichment, is provided during weekly cage cleaning to encourage natural behaviours that are crucial for maintaining the health and wellbeing of the mice. Diet and water are provided ad libitum, and environmental parameters are continuously monitored to meet legislative and welfare requirements. Mice are routinely handled using refined handling techniques, including tunnel handling and cupping, to minimise stress and ensure high standards of animal welfare. |

Note that full information on the approval of the study protocol must also be provided in the manuscript.

## Plants

**Seed stocks**

*Report on the source of all seed stocks or other plant material used. If applicable, state the seed stock centre and catalogue number. If plant specimens were collected from the field, describe the collection location, date and sampling procedures.*

**Novel plant genotypes**

*Describe the methods by which all novel plant genotypes were produced. This includes those generated by transgenic approaches, gene editing, chemical/radiation-based mutagenesis and hybridization. For transgenic lines, describe the transformation method, the number of independent lines analyzed and the generation upon which experiments were performed. For gene-edited lines, describe the editor used, the endogenous sequence targeted for editing, the targeting guide RNA sequence (if applicable) and how the editor was applied.*

**Authentication**

*Describe any authentication procedures for each seed stock used or novel genotype generated. Describe any experiments used to assess the effect of a mutation and, where applicable, how potential secondary effects (e.g. second site T-DNA insertions, mosiacism, off-target gene editing) were examined.*

