## [Peer Review File · Nature Microbiology]

Higher stability of novel live-attenuated oral poliovirus type 2 (nOPV2) despite emergence of a neurovirulent double recombinant strain in Uganda

Corresponding Author: Dr Javier Martin

Version 0:

Reviewer comments:

Reviewer #1

(Remarks to the Author)

The Authors present a thorough investigation of usage of nOPV2 in response to a vaccine-derived poliovirus type 2 outbreak in Uganda in 2022 and 2023. Whole-genome sequencing was used to characterise all nOPV2 isolates from cases of acute flaccid paralysis and environmental surveillance. The identification of a double recombinant that lost key modifications associated with the genetic stability of nOPV2 but did not lead to emergence of vaccine-derived poliovirus, was an important finding.

The work is clearly presented but the Authors are requested to consider the following comments and recommendations.

Lines 61-62. The abbreviation "5'NTR" is used throughout the manuscript but was first written in full as "5' untranslated region", which is the abbreviation for "5' UTR". In the schematic diagram of the poliovirus genome for Figure 1 and Supplementary Fig. 1, the abbreviation "5' NCR" was used, which is associated with "non-coding region". The Authors are requested to correct the initial usage to "5' non-translated region", and to ensure the abbreviation "5' NTR" is used throughout for consistency.

Line 128. What temperature was used to heat inactivate the polioviruses before they were adsorbed onto the FTA cards?

Line 151. Are the Authors able to provide the PCR primer details and amplification conditions for the custom-made tiled PCR as part of the publication, rather than upon request?

Lines 330-331. Three key mutations specified in Figure 3 and Table 1, list the mutated amino acid as "X". Namely, VP3-H77X, VP1-I143X and 2A-N36X. The Authors should clarify what "X" refers to, and state the alternative amino acid one-letter codes, if more than one mutation was identified. For example, VP1-I143T/V, or whatever amino acids replaced isoleucine.

Lines 370-373. The Authors are requested to provide a brief description of the genetic modifications identified in the two clusters of possible short-term transmission, compared to nOPV2, such as the number of mutations in VP1 and any recombination events.

Line 443. Can the Authors indicate what was Uganda's "historically general high population immunity" or OPV and IPV coverage? Since high rates of polio vaccination are considered a factor that prevented transmission of the nOPV2 double recombinant in Uganda, this will help to compare with emergence of cVDPV2-n in the countries listed in references 28 to 30.

Figure 4. For clarity, the Authors are requested to consider using the full abbreviation for the non-polio enterovirus types identified from the Kisenyi Sewage Ponds, as recommended in the publication <https://doi.org/10.1007/s00705-019-04520-6>. For example, CVA1 rather than A1 and EV-B82 rather than B82. The Authors may also consider prefacing the grouping of enteroviruses by species in the legend for each sample, by including (EV-C), (EV-A) and (EV-B) in front of the respective species. For example, for UGA_22_001, (EV-C) CVA1, CVA13, CVA17, CVA19, CVA24, EV-C116, EV-C99; (EV-A) CVA7, EV-A76 etc.

Table 1. Correct E1295K listed under possible effect for VP1-N171D, should be described as VP1-E295K, not E1295K.

Grammar

Line 306. Amend to "...category 2 as they show a double recombinant structure..."

Figure 2. Amend to "Number of mutations from the nOPV2 vaccine..." rather than "Numbers..."

Bruce Thorley

Reviewer #2

(Remarks to the Author)

Tushabe et al. performed genomic surveillance of polio virus after two nationwide novel oral poliovirus vaccine type 2 campaigns in Uganda. The authors used whole genome sequencing to characterize 233 isolates from stool and sewage samples. Within these samples, a variety of genetic changes was found, ranging from no modifications to a double recombinant strain that lost all key modifications that were included in the novel vaccine strain. This recombinant strain was further experimentally characterized in mice and found to have increased neurovirulence. Despite this increased neurovirulence, the recombinant strain was only found in sewage samples, without evidence for widespread transmission. Overall, this study

provides important insights in the importance of genomic surveillance to monitor campaigns aimed at polio eradication.

Major comments

1. Sampling strategy of stool and sewage samples is a key component of this study that is critical to the interpretation of the findings, but a description is currently missing in the methods. Particularly, it is unclear whether stool sampling was conducted among both vaccinated and unvaccinated individuals (as well as other demographics of sampled individuals). This is an important distinction to understand whether positive samples were the result of (prolonged) shedding of virus after vaccination or evidence of transmission of vaccine-derived virus to unvaccinated individuals.
2. Lines 288-307: This paragraph contains important information on observed mutation profiles and their frequencies. Given the importance of these data for this study, I would recommend including these categories, mutations profiles, changes, and frequencies as a main text table.
3. To ensure reproducibility of the work, the authors should include primer sequences as part of the custom-made tiled PCR approach (lines 149-151) publicly available. Raw sequencing data should also be deposited in a public repository, such as SRA (particularly sequences for the recombinant strain to allow reproducibility/verification of findings). Additionally, metadata and consensus genomes generated as part of this study should also be included in the manuscript instead of only being available upon request. Currently, consensus genomes are not available (yet) on GenBank. As a result, I'm not able to review the genomic data generated as part of this study.
4. Lines 192-196: Additional description is needed to explain how consensus genomes were generated for recombinant strains, from Illumina and Nanopore data. If this was done using a reference-based alignment, then which reference genomes were used and how was accurate aligned against multiple viruses (polio and enteroviruses) ensured?
5. The title is a bit misleading as the study is not specifically focused on virus evolution. I would suggest an alternative title that highlights this was a genomic surveillance study. For example: Genomic surveillance of the novel oral poliovirus type 2 (nOPV2) vaccine strain: detection of a double recombinant strain in Uganda

Minor comments

1. The manuscript contains many abbreviations (e.g. OPV, nOPV, WPV, cVDPV, IPV, etc) and it is hard to keep track of what every abbreviation means. I would recommend minimizing the use of abbreviations, especially when abbreviations are used few times or when abbreviations are very similar.
2. Lines 46-51: The first sentence of the introduction is very long and I would suggest to split it into two to improve readability: "...for more than 60 years. This has helped.."
3. Lines 114-117: I would recommend removing these sentences as these fit better in the discussion.
4. The first paragraph of the results seems to be a better fit for the introduction, as it seems that the vaccination campaign was published as part of a prior study (ref 140).
5. Supplementary Fig. 2: Please include details on how tree was generated, what background genomes are, and what x-axis represents

Decision Letter:

18th June 2025

Dear Dr Martin,

Thank you for your patience while your manuscript "Evolution of the live-attenuated novel oral poliovirus type 2 (nOPV2) vaccine: detection of a double recombinant strain in Uganda." was under peer-review at Nature Microbiology. It has now been seen by 2 referees, whose expertise and comments you will find at the end of this email. Although they find your work of some potential interest, they have raised a number of concerns that will need to be addressed before we can consider publication of the work in Nature Microbiology.

In particular, both reviewers agree that clarifications and better reporting are needed, including making the sequences under accession numbers PV576826-PV577058 publicly available. Please address all the reviewers issues, we would be happy to look at a revised manuscript.

Please include a data availability statement as a separate section after Methods but before references, under the heading "Data Availability". This section should inform readers about the availability of the data used to support the conclusions of your study. This information includes accession codes to public repositories (data banks for protein, DNA or RNA sequences, microarray, proteomics data etc...), references to source data published alongside the paper, unique identifiers such as URLs to data repository entries, or data set DOIs, and any other statement about data availability. At a minimum, you should include the

following statement: "The data that support the findings of this study are available from the corresponding author upon request", mentioning any restrictions on availability. If DOIs are provided, we also strongly encourage including these in the Reference list (authors, title, publisher (repository name), identifier, year). For more guidance on how to write this section please see: <http://www.nature.com/authors/policies/data/data-availability-statements-data-citations.pdf>

* If you have not done so already we suggest that you begin to revise your manuscript so that it conforms to our Article format instructions at <http://www.nature.com/nmicrobiol/info/final-submission>. Refer also to any guidelines provided in this letter.

When submitting the revised version of your manuscript, please pay close attention to our [href="https://www.nature.com/nature-portfolio/editorial-policies/image-integrity">Digital Image Integrity Guidelines. and to the following points below:](https://www.nature.com/nature-portfolio/editorial-policies/image-integrity)

EXTENDED DATA FIGURES

Link Redacted

Note: This url links to your confidential homepage and associated information about manuscripts you may have submitted or be reviewing for us. If you wish to forward this e-mail to co-authors, please delete this link to your homepage first.

Nature Microbiology is committed to improving transparency in authorship. As part of our efforts in this direction, we are now requesting that all authors identified as 'corresponding author' on published papers create and link their Open Researcher and Contributor Identifier (ORCID) with their account on the Manuscript Tracking System (MTS), prior to acceptance. This applies to primary research papers only. ORCID helps the scientific community achieve unambiguous attribution of all scholarly contributions. You can create and link your ORCID from the home page of the MTS by clicking on 'Modify my Springer Nature account'. For more information please visit [please visit www.springernature.com/orcid](http://www.springernature.com/orcid).

If you wish to submit a suitably revised manuscript we would hope to receive it within 6 months. If you cannot send it within this time, please let us know. We will be happy to consider your revision, even if a similar study has been accepted for publication at Nature Microbiology or published elsewhere (up to a maximum of 6 months).

Yours sincerely,

Reviewer Expertise:

Referee #1: Polio, polio vaccines
Referee #2: Epidemiology, phylogenetics

Reviewer Comments:

Reviewer #1 (Remarks to the Author):

The Authors present a thorough investigation of usage of nOPV2 in response to a vaccine-derived poliovirus type 2 outbreak in Uganda in 2022 and 2023. Whole-genome sequencing was used to characterise all nOPV2 isolates from cases of acute flaccid paralysis and environmental surveillance. The identification of a double recombinant that lost key modifications associated with the genetic stability of nOPV2 but did not lead to emergence of vaccine-derived poliovirus, was an important finding.

The work is clearly presented but the Authors are requested to consider the following comments and recommendations.

Lines 61-62. The abbreviation "5'NTR" is used throughout the manuscript but was first written in full as "5' untranslated region", which is the abbreviation for "5' UTR". In the schematic diagram of the poliovirus genome for Figure 1 and Supplementary Fig. 1, the abbreviation "5' NCR" was used, which is associated with "non-coding region". The Authors are requested to correct the initial usage to "5' non-translated region", and to ensure the abbreviation "5' NTR" is used throughout for consistency.

Line 128. What temperature was used to heat inactivate the polioviruses before they were adsorbed onto the FTA cards?

Line 151. Are the Authors able to provide the PCR primer details and amplification conditions for the custom-made tiled PCR as part of the publication, rather than upon request?

Lines 330-331. Three key mutations specified in Figure 3 and Table 1, list the mutated amino acid as "X". Namely, VP3-H77X, VP1-I143X and 2A-N36X. The Authors should clarify what "X" refers to, and state the alternative amino acid one-letter codes, if more than one mutation was identified. For example, VP1-I143T/V, or whatever amino acids replaced isoleucine.

Lines 370-373. The Authors are requested to provide a brief description of the genetic modifications identified in the two clusters of possible short-term transmission, compared to nOPV2, such as the number of mutations in VP1 and any recombination events.

Line 443. Can the Authors indicate what was Uganda's "historically general high population immunity" or OPV and IPV coverage? Since high rates of polio vaccination are considered a factor that prevented transmission of the nOPV2 double recombinant in Uganda, this will help to compare with emergence of cVDPV2-n in the countries listed in references 28 to 30.

Figure 4. For clarity, the Authors are requested to consider using the full abbreviation for the non-polio enterovirus types identified from the Kisenyi Sewage Ponds, as recommended in the publication <https://doi.org/10.1007/s00705-019-04520-6>. For example, CVA1 rather than A1 and EV-B82 rather than B82. The Authors may also consider prefacing the grouping of enteroviruses by species in the legend for each sample, by including (EV-C), (EV-A) and (EV-B) in front of the respective species. For example, for UGA_22_001, (EV-C) CVA1, CVA13, CVA17, CVA19, CVA24, EV-C116, EV-C99; (EV-A) CVA7, EV-A76 etc.

Table 1. Correct E1295K listed under possible effect for VP1-N171D, should be described as VP1-E295K, not E1295K.

Grammar

Line 306. Amend to "...category 2 as they show a double recombinant structure..."

Figure 2. Amend to "Number of mutations from the nOPV2 vaccine..." rather than "Numbers..."

Bruce Thorley

Reviewer #2 (Remarks to the Author):

Tushabe et al. performed genomic surveillance of polio virus after two nationwide novel oral poliovirus vaccine type 2 campaigns in Uganda. The authors used whole genome sequencing to characterize 233 isolates from stool and sewage samples. Within these samples, a variety of genetic changes was found, ranging from no modifications to a double recombinant strain that lost all key modifications that were included in the novel vaccine strain. This recombinant strain was further experimentally characterized in mice and found to have increased neurovirulence. Despite this increased neurovirulence, the recombinant strain was only found in sewage samples, without evidence for widespread transmission. Overall, this study provides important insights in the importance of genomic surveillance to monitor campaigns aimed at polio eradication.

Major comments

1. Sampling strategy of stool and sewage samples is a key component of this study that is critical to the interpretation of the findings, but a description is currently missing in the methods. Particularly, it is unclear whether stool sampling was conducted among both vaccinated and unvaccinated individuals (as well as other demographics of sampled individuals). This is an important distinction to understand whether positive samples were the result of (prolonged) shedding of virus after vaccination or evidence of transmission of vaccine-derived virus to unvaccinated individuals.

2. Lines 288-307: This paragraph contains important information on observed mutation profiles and their frequencies. Given the importance of these data for this study, I would recommend including these categories, mutations profiles, changes, and frequencies as a main text table.

3. To ensure reproducibility of the work, the authors should include primer sequences as part of the custom-made tiled PCR approach (lines 149-151) publicly available. Raw sequencing data should also be deposited in a public repository, such as SRA (particularly sequences for the recombinant strain to allow reproducibility/verification of findings). Additionally, metadata and consensus genomes generated as part of this study should also be included in the manuscript instead of only being available upon request. Currently, consensus genomes are not available (yet) on GenBank. As a result, I'm not able to review the genomic data generated as part of this study.

4. Lines 192-196: Additional description is needed to explain how consensus genomes were generated for recombinant strains, from Illumina and Nanopore data. If this was done using a reference-based alignment, then which reference genomes were used and how was accurate aligned against multiple viruses (polio and enteroviruses) ensured?

5. The title is a bit misleading as the study is not specifically focused on virus evolution. I would suggest an alternative title that highlights this was a genomic surveillance study. For example: Genomic surveillance of the novel oral poliovirus type 2 (nOPV2) vaccine strain: detection of a double recombinant strain in Uganda

Minor comments

1. The manuscript contains many abbreviations (e.g. OPV, nOPV, WPV, cVDPV, IPV, etc) and it is hard to keep track of what every abbreviation means. I would recommend minimizing the use of abbreviations, especially when abbreviations are used few times or when abbreviations are very similar.
2. Lines 46-51: The first sentence of the introduction is very long and I would suggest to split it into two to improve readability: “..for more than 60 years. This has helped..”
3. Lines 114-117: I would recommend removing these sentences as these fit better in the discussion.
4. The first paragraph of the results seems to be a better fit for the introduction, as it seems that the vaccination campaign was published as part of a prior study (ref 140).
5. Supplementary Fig. 2: Please include details on how tree was generated, what background genomes are, and what x-axis represents

Version 1:

Reviewer comments:

Reviewer #1

(Remarks to the Author)

The Reviewer thanks the Authors for considering the comments and recommendations submitted to them and considers they were addressed satisfactorily.

Reviewer #2

(Remarks to the Author)

I would like to thank the authors for revising the manuscript. I have two remaining comments that require clarification:

Lines 213-214: Please clarify that primer sequences were trimmed when processing sequencing data from tiled amplicon sequencing and include the threshold (depth of coverage) for calling consensus genomes. Genome completeness and depth of coverage across the genome for each sample should be included in supplementary table 3.

Line 243: This sentence is missing references: “Sequence contigs were built by reference-guided assembly and/or de novo assembly as extensively described before (refs).”

Decision Letter:

Our ref: NMICROBIOL-25041275A

9th October 2025

Dear Dr. Martin,

Thank you for submitting your revised manuscript "Evolution of the live-attenuated novel oral poliovirus type 2 (nOPV2) vaccine: detection of a double recombinant strain in Uganda." (NMICROBIOL-25041275A). It has now been seen by the original referees and their comments are below. The reviewers find that the paper has improved in revision, and therefore we'll be happy in principle to publish it in Nature Microbiology, pending minor revisions to satisfy the referees' final requests and to comply with our editorial and formatting guidelines.

Thank you again for your interest in Nature Microbiology Please do not hesitate to contact me if you have any questions.

Sincerely,

Reviewer #1 (Remarks to the Author):

The Reviewer thanks the Authors for considering the comments and recommendations submitted to them and considers they were addressed satisfactorily.

Reviewer #2 (Remarks to the Author):

I would like to thank the authors for revising the manuscript. I have two remaining comments that require clarification:

Lines 213-214: Please clarify that primer sequences were trimmed when processing sequencing data from tiled amplicon sequencing and include the threshold (depth of coverage) for calling consensus genomes. Genome completeness and depth of coverage across the genome for each sample should be included in supplementary table 3.

Line 243: This sentence is missing references: "Sequence contigs were built by reference-guided assembly and/or de novo assembly as extensively described before (refs)."

Version 2:

Decision Letter:

12th November 2025

Dear Dr Martin,

I am pleased to accept your Article "Evolution of the live-attenuated novel oral poliovirus type 2 (nOPV2) vaccine: detection of a double recombinant strain in Uganda." for publication in Nature Microbiology. Thank you for having chosen to submit your work to us and many congratulations.

Authors may need to take specific actions to achieve compliance with funder and institutional open access mandates. If your research is supported by a funder that requires immediate open access (e.g. according to [Plan S principles](https://www.springernature.com/gp/open-science/plan-s-compliance) or the [NIH public access policy](https://www.springernature.com/gp/open-science/us-federal-agency-compliance)) then you should select the gold OA route, and we will direct you to the compliant route where possible. Because authors warrant under our subscription licensing terms that they haven't committed to licensing any version of their article under a licence inconsistent with the terms of our agreement – including the applicable embargo period – publication under the subscription model isn't suitable for authors whose funders require no embargo.

With kind regards,

P.S. Click on the following link if you would like to recommend Nature Microbiology to your librarian
<http://www.nature.com/subscriptions/recommend.html#forms>

** Visit the Springer Nature Editorial and Publishing website at http://editorial-jobs.springernature.com?utm_source=ejP_NMicro_email&utm_medium=ejP_NMicro_email&utm_campaign=ejp_NMicro for more information about our career opportunities. If you have any questions please click [here](mailto:editorial.publishing.jobs@springernature.com).

Response to Reviewers

Manuscript ID: NMICROBIOL-25041275

Title: Evolution of the live-attenuated novel oral poliovirus type 2 (nOPV2) vaccine: detection of a double recombinant strain in Uganda.

Dear Editor,

We thank you and the reviewers for the careful evaluation of our manuscript and for the constructive comments provided. We have revised the text accordingly and believe that these changes have substantially improved the clarity and rigor of the paper. Below we provide a detailed, point-by-point response to each comment. Reviewer comments are shown in *italics*, followed by our responses. All textual changes have been incorporated into the revised manuscript.

Reviewer #1

Lines 61–62. The abbreviation “5’NTR” is used throughout the manuscript but was first written in full as “5’ untranslated region”, which is the abbreviation for “5’ UTR”. In the schematic diagram of the poliovirus genome for Figure 1 and Supplementary Fig. 1, the abbreviation “5’ NCR” was used, which is associated with “non-coding region”. The Authors are requested to correct the initial usage to “5’ non-translated region”, and to ensure the abbreviation “5’ NTR” is used throughout for consistency.

Response: We have corrected the terminology throughout the manuscript to consistently use “5’NTR,” as recommended.

Line 128. What temperature was used to heat inactivate the polioviruses before they were adsorbed onto the FTA cards?

Response: We inactivated poliovirus isolates by spotting supernatant onto FTA cards (Whatman), which contain chemicals that inactivate virus while preserving RNA. This GPLN-standard method includes a 70 °C for 4 min heat treatment before spotting. The description has been added to the *Sampling, virus isolation, intratypic differentiation (ITD) and shipping of poliovirus materials* section in *Methods*.

Line 151. Are the Authors able to provide the PCR primer details and amplification conditions for the custom-made tiled PCR as part of the publication, rather than upon request?

Response: Yes. The custom tiled PCR protocol has been published and is accessible online and primer sequences are now included in Supplementary Table 1. The *Sampling, virus isolation, intratypic differentiation (ITD) and shipping of poliovirus materials* section in *Methods* has been updated accordingly.

Lines 330-331. Three key mutations specified in Figure 3 and Table 1, list the mutated amino acid as “X”. Namely, VP3-H77X, VP1-I143X and 2A-N36X. The Authors should clarify what “X” refers to, and state the alternative amino acid one-letter codes, if more than one mutation was identified. For example, VP1-I143T/V, or

whatever amino acids replaced isoleucine.

Response: All amino acid substitutions are now fully specified in Figure 3 and Table 1.

Lines 370-373. The Authors are requested to provide a brief description of the genetic modifications identified in the two clusters of possible short-term transmission, compared to nOPV2, such as the number of mutations in VP1 and any recombination events.

Response: We added text describing genetic linkage between isolates and revised the tree to highlight only closely related isolates using GenBank identifiers. No recombination events were identified.

Line 443. Can the Authors indicate what was Uganda's "historically general high population immunity" or OPV and IPV coverage? Since high rates of polio vaccination are considered a factor that prevented transmission of the nOPV2 double recombinant in Uganda, this will help to compare with emergence of cVDPV2-n in the countries listed in references 28 to 30.

Response: According to WHO/UNICEF Estimates of National Immunization Coverage, Uganda has consistently maintained coverage above 80% for both OPV3 and IPV over the years. In contrast, countries like Nigeria, where transmission of recombinant polioviruses has been reported have had estimates for OPV3 and IPV consistently below 70% (WHO Immunization Data portal - African Region). This information has been added to the Discussion section. Line 589 to 595.

Figure 4. For clarity, the Authors are requested to consider using the full abbreviation for the non-polio enterovirus types identified from the Kisenyi Sewage Ponds, as recommended in the publication <https://doi.org/10.1007/s00705-019-04520-6>. For example, CVA1 rather than A1 and EV-B82 rather than B82. The Authors may also consider prefacing the grouping of enteroviruses by species in the legend for each sample, by including (EV-C), (EV-A) and (EV-B) in front of the respective species. For example, for UGA_22_001, (EV-C) CVA1, CVA13, CVA17, CVA19, CVA24, EV-C116, EV-C99; (EV-A) CVA7, EV-A76 etc.

Response: Non-polio enteroviruses are now listed with full designations (e.g., CVA1, EV-B82) in both figure and legend. However, we have maintained the legend as before as it is automatically generated by Excel.

Table 1. Correct E1295K listed under possible effect for VP1-N171D, should be described as VP1-E295K, not E1295K.

Response: Corrected to VP1-E295K.

Grammar

Line 306. Amend to "...category 2 as they show a double recombinant structure..."

Figure 2. Amend to “Number of mutations from the nOPV2 vaccine...” rather than “Numbers...”

Response: Both corrections have been implemented.

Reviewer #2

Tushabe et al. performed genomic surveillance of polio virus after two nationwide novel oral poliovirus vaccine type 2 campaigns in Uganda. The authors used whole genome sequencing to characterize 233 isolates from stool and sewage samples. Within these samples, a variety of genetic changes was found, ranging from no modifications to a double recombinant strain that lost all key modifications that were included in the novel vaccine strain. This recombinant strain was further experimentally characterized in mice and found to have increased neurovirulence. Despite this increased neurovirulence, the recombinant strain was only found in sewage samples, without evidence for widespread transmission. Overall, this study provides important insights in the importance of genomic surveillance to monitor campaigns aimed at polio eradication.

Major comments

1. Sampling strategy of stool and sewage samples is a key component of this study that is critical to the interpretation of the findings, but a description is currently missing in the methods. Particularly, it is unclear whether stool sampling was conducted among both vaccinated and unvaccinated individuals (as well as other demographics of sampled individuals). This is an important distinction to understand whether positive samples were the result of (prolonged) shedding of virus after vaccination or evidence of transmission of vaccine-derived virus to unvaccinated individuals.

Response: Stool samples were collected through nationwide AFP surveillance, where children <15 years with symptoms of weakness or paralysis are investigated regardless of vaccination status. Two stool samples were collected 24–48 h apart and transported under cold chain. Sewage grab samples were collected regularly from 11 environmental surveillance sites. These collections followed nOPV2 campaigns targeting all children <5 years so it is likely that most children of any age were exposed to the vaccine one way or another. Seven of 84 children sampled were confirmed vaccinees, but transmission within households likely accounts for additional isolates. Sequence analysis revealed no sustained transmission, and the maximum interval between vaccination and a positive stool sample was 46 days, excluding prolonged shedding. Details have been added to *Methods and Results* sections.

2. Lines 288-307: This paragraph contains important information on observed mutation profiles and their frequencies. Given the importance of these data for this study, I would recommend including these categories, mutations profiles, changes, and frequencies as a main text table.

Response: A new Table 1 has been added summarizing mutation categories, profiles, and frequencies. In addition, the former Supplementary Table 1 (now Supplementary Table 2) has been revised for clarity, and it now shows only mutation combinations at sites with evidence suggesting a potential effect in reducing attenuation.

3. To ensure reproducibility of the work, the authors should include primer sequences as part of the custom-made tiled PCR approach (lines 149-151) publicly available. Raw sequencing data should also be deposited in a public repository, such as SRA (particularly sequences for the recombinant strain to allow reproducibility/verification of findings). Additionally, metadata and consensus genomes generated as part of this study should also be included in the manuscript instead of only being available upon request. Currently, consensus genomes are not available (yet) on GenBank. As a result, I'm not able to review the genomic data generated as part of this study.

Response:

- The custom tiled PCR protocol has been published and is accessible online and primer sequences are now included in Supplementary Table 1. The *Sampling, virus isolation, intratypic differentiation (ITD) and shipping of poliovirus materials* section in *Methods* has been updated accordingly.
- All consensus sequences are now fully accessible from GenBank under accession numbers PV576826-PV577058.
- Raw fastq NGS files generated in this study corresponding to double recombinant viruses and enterovirus sequences from sewage have been deposited in the NCBI Sequence Read Archive under project code Bio project number: PRJNA1295797.

All this information has been added to relevant sections in the manuscript.

4. Lines 192-196: Additional description is needed to explain how consensus genomes were generated for recombinant strains, from Illumina and Nanopore data. If this was done using a reference-based alignment, then which reference genomes were used and how was accurate aligned against multiple viruses (polio and enteroviruses) ensured?

Response: Response: fastq files from both Nanopore and Illumina sequencing were analysed using Geneious 10.2.3 software. Sequence contigs were built by reference-guided assembly and/or de novo assembly as extensively described before. Raw sequence data were imported into Geneious 10.2.6 and sequence contigs were built by reference-guided assembly. Fastq reads were initially mapped simultaneously to Sabin 1, nOPV2 and Sabin 3 whole-genome sequences and contig sequences were generated. Potential nOPV2 recombinant sequences were identified in contigs showing low or no sequence coverage outside the capsid coding region. Filtered reads were then iteratively reassembled to consensus sequences covering the nOPV2 capsid region using stringent assembly conditions to build whole-genome

contig sequences (Illumina: ≥ 50 bp overlap, $\geq 98\%$ identity; Nanopore: ≥ 2000 bp overlap, $\geq 95\%$ identity). Final consensus sequences were obtained by assigning the most common nucleotide sequence to each nucleotide position within each contig. When necessary and to confirm the results, fastq reads were also independently assembled de novo, using the same stringent assembly conditions. For whole-genome PCR products, Nanopore sequencing assembly required only single-step mapping to Sabin 1, nOPV2, and Sabin 3. Results using different PCR amplification strategies and assembly approaches yielded consistent results. Manual analyses for visualizing and quantifying assembly results were performed throughout the process. Final consensus sequences are available from GenBank under accession numbers PV576826-PV577058. All the information above has been added to the *Methods* section.

5. *The title is a bit misleading as the study is not specifically focused on virus evolution. I would suggest an alternative title that highlights this was a genomic surveillance study. For example: Genomic surveillance of the novel oral poliovirus type 2 (nOPV2) vaccine strain: detection of a double recombinant strain in Uganda*

Response: While we appreciate the suggestion, we believe the current title accurately reflects the evolutionary processes observed in nOPV2 during replication in humans. We therefore prefer to retain the original title.

Minor comments

1. The manuscript contains many abbreviations (e.g. OPV, nOPV, WPV, cVDPV, IPV, etc) and it is hard to keep track of what every abbreviation means. I would recommend minimizing the use of abbreviations, especially when abbreviations are used few times or when abbreviations are very similar.

Response: Widely used abbreviations (OPV, nOPV, cVDPV) have been retained, but less common ones (e.g., WPV, GPLN, IPV) have been minimized.

2. Lines 46-51: The first sentence of the introduction is very long and I would suggest to split it into two to improve readability: “..for more than 60 years. This has helped..”

Response: The first sentence of the Introduction has been split for clarity, as suggested.

3. Lines 114-117: I would recommend removing these sentences as these fit better in the discussion

Response: We have shortened the text but retained a brief mention at the end of the Introduction to emphasize this key finding for the reader.

4. The first paragraph of the results seems to be a better fit for the introduction, as it seems that the vaccination campaign was published as part of a prior study (ref 140).

Response: The first paragraph of the Results has been moved to the Introduction, as recommended.

5. Supplementary Fig. 2: Please include details on how tree was generated, what background genomes are, and what x-axis represents.

Response: Details on tree construction, background genomes, and the x-axis have been added to the legend.

We are grateful to the reviewers for their constructive feedback. Their comments have helped us improve the clarity, reproducibility, and presentation of our work. We hope the revised manuscript addresses all concerns satisfactorily, and we look forward to your further consideration.

Sincerely,
Javier Martin
on behalf of all authors

Response to Reviewers

Manuscript ID: NMICROBIOL-25041275

Title: Higher stability of novel live-attenuated oral poliovirus type 2 (nOPV2) despite emergence of a neurovirulent double recombinant strain in Uganda.

Dear Editor,

We thank you and the reviewers for the careful evaluation of our manuscript and for the constructive comments provided. We have revised the text accordingly and believe that these changes have substantially improved the clarity and rigor of the paper. Below we provide a detailed, point-by-point response to each comment. Reviewer comments are shown in *italics*, followed by our responses. All textual changes have been incorporated into the revised manuscript and minor typographic errors have been corrected as required.

Reviewer #2:

Remarks to the Author:

I would like to thank the authors for revising the manuscript. I have two remaining comments that require clarification:

Lines 213-214: Please clarify that primer sequences were trimmed when processing sequencing data from tiled amplicon sequencing and include the threshold (depth of coverage) for calling consensus genomes. Genome completeness and depth of coverage across the genome for each sample should be included in supplementary table 3.

Response:

We have modified the manuscript as advised to include this information. Supplementary Table 3 (now Extended Data Table 3) has been updated accordingly.

Line 243: This sentence is missing references: "Sequence contigs were built by reference-guided assembly and/or de novo assembly as extensively described before (refs)."

Response:

This has been corrected as advised to make sure the relevant references were included in the text.

We are grateful to the reviewers for their constructive feedback. Their comments have helped us improve the clarity, reproducibility, and presentation of our work. We hope the revised manuscript addresses all concerns satisfactorily, and we look forward to your further consideration.

Sincerely,

Javier Martin

on behalf of all authors